# Trust Functions: Near-Lossless Weak-to-Strong Generalization by Learning When to Trust the Weak Teacher

**Arda Uzunoğlu** [* 1]   **Alvin Zhang** [* 1]   **Daniel Khashabi** [1]

 Code      Website

## Abstract

Weak-to-strong generalization studies how to improve a strong student using supervision from a weaker teacher when reliable labels are scarce. We view this primarily as a data selection problem, where the key challenge is to identify which weak labels are reliable enough to serve as a training signal. To address this, we introduce *trust functions* that assign each weak label a scalar *trust* score and use these scores to filter weak supervision. Across several domains, including world knowledge, quantitative reasoning, and strategy games, trust filtering yields students that match and sometimes surpass ground-truth supervision, **achieving near-lossless weak-to-strong generalization**. Moreover, trust functions enable an iterative *weak-to-strong chain* that compounds gains by training a student and reusing it as the next teacher, amplifying the gains. There are several mechanisms to which advantage of trust functions can be attributed. They retain data points that induce an implicit easy-first curriculum, often recover optimal labels where ground truth labels are suboptimal, and produce more aligned gradients.

## 1. Introduction

As large language models (LLMs) approach and sometimes surpass human performance on complex tasks (Brodeur et al., 2025), the traditional assumption that humans can provide reliable supervision breaks down. This shift motivates weak-to-strong generalization, where a strong student is improved using supervision from a weaker teacher

in the absence of reliable supervision (Burns et al., 2023). Empirically, prior work shows that weak supervision can produce students that surpass their teachers, but it fails to close the gap to training on ground-truth labels (Burns et al., 2023; Agrawal et al., 2024; Lang et al., 2025). This gap is attributed to the imperfect training labels (weak label) generated by the weak teacher, which can (i) propagate weak label errors unless the data geometry allows the strong model to reliably correct them (Lang et al., 2024) and (ii) omit task-relevant directions that lie outside the weak teacher's representational span (Xue et al., 2025). These errors are often amplified under distribution shift, leading weak supervision to destabilize learning (Yu et al., 2021).

Mitigating such errors admits multiple perspectives. Consequently, weak-to-strong generalization can be approached along several axes, including optimization (Burns et al., 2023), learning algorithms (Liu & Alahi, 2024), and data selection (Agrawal et al., 2024). In this work, we focus on the *data selection* axis and aim to identify the subset of weakly labeled examples whose supervision most effectively improves the student, holding the architecture and training algorithm fixed. We formalize this decision via an umbrella notion of *trust functions*, which map a weak label to a scalar *trust score*, indicating its estimated accuracy, thus its expected usefulness in training. Existing methods typically instantiate trust using output-level heuristics such as entropy (Kuhn et al., 2023), inter-model agreement (Lang et al., 2025), or explicit self-evaluation (Tian et al., 2023). While effective in some regimes, these signals can be poorly calibrated on complex tasks and brittle under distribution shift, often assigning high trust to confident errors and low trust to correct but uncertain solutions (Tian et al., 2023; Fadeeva et al., 2023).

Motivated by this weakness, we move beyond output-level heuristics and instead base trust on internal representations. We introduce *neural trust functions* that estimate a weak label's correctness using features from the teacher's internal activations, leveraging evidence that intermediate representations contain separable signals of answer correctness that may be obscured in the final decoded output (Kadavath

*Equal contribution ; alphabetical by last name [1]Department of Computer Science, Johns Hopkins University, Maryland, United States. Correspondence to: Arda Uzunoğlu <auzunog1@jh.edu>, Alvin Zhang <bzhang90@jh.edu>.

*Proceedings of the $43^{rd}$ International Conference on Machine Learning*, Seoul, South Korea. PMLR 306, 2026. Copyright 2026 by the author(s).

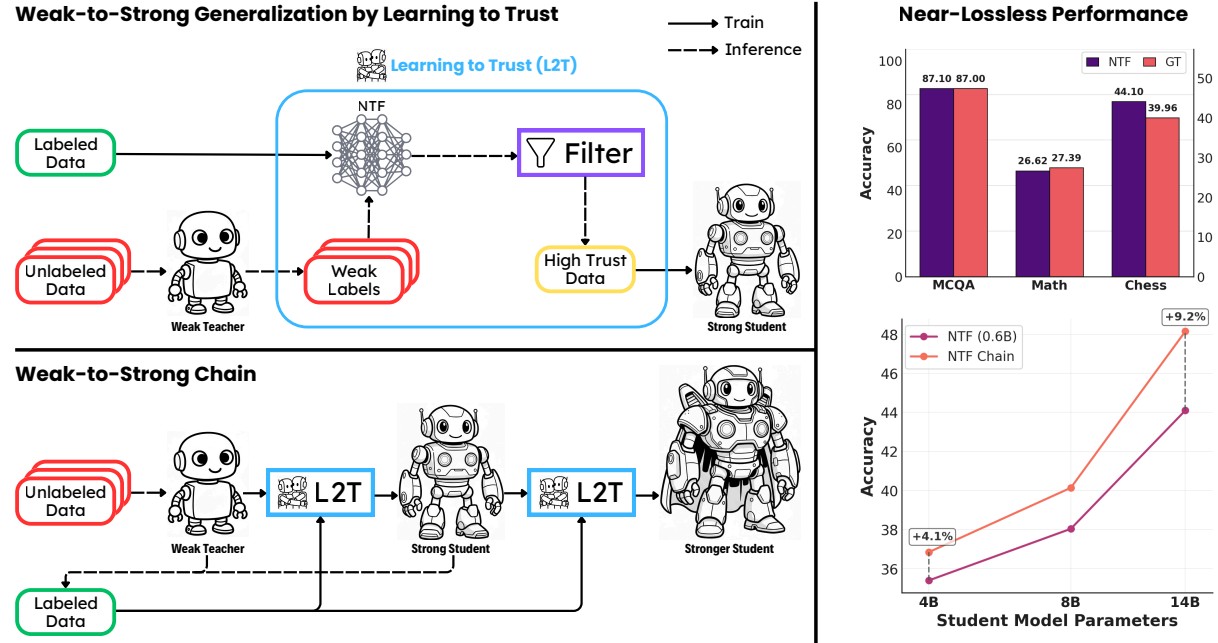

*Figure 1.* **(1) Learning to Trust (top-left):** Given a small labeled set, we train a neural trust function (NTF) to predict whether weak labels are reliable. We then use the NTF to filter weak labels to produce a high-trust subset, which is used to train the strong student model. **(2) Weak-to-Strong Chain (bottom-left):** This procedure can be applied iteratively across multiple generations of students, forming a weak-to-strong chain that compounds gains and outperforms direct weak-to-strong transfer as students scale. **(3) Results (right):** Across multiple domains, L2T achieves near-lossless performance compared to training on ground-truth (GT) labels (top). Moreover, chaining yields *snowballing gains* across the iterations (bottom).

et al., 2022; Kuhn et al., 2023). We train these neural trust functions on a labeled source distribution and deploy them zero-shot under *in-domain distribution shift* (§3.2) to select weak labels on an unlabeled target distribution. Across *world knowledge* (MCQA), *quantitative reasoning* (mathematical problems), and *strategy games* (chess puzzles), students trained on our trust-filtered weak labels achieve near-lossless, ground-truth-level performance, and in some cases improve upon it. Furthermore, trust filtering enables an iterative *weak-to-strong chain* that reuses each trained student as the next teacher, yielding *snowballing* improvements across iterations. Our contributions are as follows:

- We formalize *trust functions* for weak-to-strong generalization and instantiate them as *neural* predictors of weak label correctness using teacher hidden states that generalize under *in-domain distribution shift*.

- We demonstrate that, across *world knowledge*, *quantitative reasoning*, and *strategy games*, trust-filtered weak supervision training matches and sometimes exceeds ground-truth supervision training.

- We propose an iterative weak-to-strong *chain* enabled by trust-filtered supervision, yielding compounding gains across successive student-teacher iterations.

## 2. Trust Functions

We formalize *trust functions* as scoring functions that estimate the reliability of weak labels.

### 2.1. Preliminaries

**Notation.** Let $\mathcal{D} = \{(x_i, y_i)\}_{i=1}^{N}$ be a dataset of input–output pairs with $x_i \in \mathcal{X}$ and $y_i \in \mathcal{Y}$. Given a weak teacher $\pi_{\mathcal{W}} : \mathcal{X} \to \hat{\mathcal{Y}}$, we denote its output by $\hat{y}_i = \pi_{\mathcal{W}}(x_i)$.

**Trust functions.** A *trust score* is a scalar assigned to $\hat{y}_i$ that measures its reliability as supervision, which we interpret as the estimated probability that $\hat{y}_i$ is correct. Let $g_{\pi_{\mathcal{W}}} : \mathcal{X} \times \hat{\mathcal{Y}} \to \mathcal{G}$ be a feature extractor applied to the input-prediction pair $(x_i, \hat{y}_i)$. A *trust function* is any mapping $\tau : \mathcal{G} \to [0, 1]$ that produces a trust score $t_i = \tau(g_{\pi_{\mathcal{W}}}(x_i, \hat{y}_i))$. Larger $t_i$ indicates higher trust and is favorable (i.e., $\hat{y}_i$ is more likely to be correct). Importantly, $t_i$ must be computable without access to the ground-truth label $y_i$ at inference time.

### 2.2. Neural Trust Functions

We focus on *neural trust functions*, where a neural network $\tau$ predicts the correctness of a weak label generated by $\pi_{\mathcal{W}}$ from features of the pair $(x, \hat{y})$. We train $\tau$ on a labeled source dataset and apply it to unlabeled target data to produce trust scores. While trust scores can be used for filtering

and loss reweighting, we focus on *filtering* in this work.

**Input.** In our instantiation, $g_{\pi_\mathcal{W}}(x, \hat{y})$ is the last-layer hidden state corresponding to the final generated token. Because this token attends to the full prefix (input and any intermediate reasoning), it provides a compact summary for predicting correctness. We use a single-token representation (no pooling) by default, and ablate layer choice, token position, and pooling strategies in App. A.1.

**Architecture.** We parameterize $\tau$ as a residual (He et al., 2015) MLP with RMSNorm-SwiGLU (Shazeer, 2020) blocks, Dropout (Srivastava et al., 2014), and stochastic depth (Huang et al., 2016), followed by a final RMSNorm (Zhang & Sennrich, 2019) and a linear head producing a scalar logit. We convert logits to trust scores with a sigmoid. We train with mini-batch AdamW (Loshchilov & Hutter, 2019) (with weight decay) using a class-reweighted binary cross-entropy loss to account for label imbalance.

**Computational Cost.** Let $\mathcal{D}_\ell$ be the labeled source set and $\mathcal{D}_u$ the unlabeled target pool. Let $\bar{C}_{\text{teacher}}$ denote the average cost of one weak-teacher forward pass, including both weak-label generation and hidden-state extraction. Let $d$ be the teacher hidden dimension, $w$ the NTF hidden width, $b$ the number of NTF layers, and $e$ the number of NTF training epochs. We denote by $C_{\text{NTF}} = O(dw + bw^2)$ the per-example cost of one NTF update or scoring pass. The total cost of generating weak labels, training the NTF on the labeled source set, and applying it to the unlabeled target pool is therefore

$$C_{\text{total}} = O\big(\bar{C}_{\text{teacher}} \cdot (|\mathcal{D}_\ell| + |\mathcal{D}_u|) + C_{\text{NTF}} \cdot (e|\mathcal{D}_\ell| + |\mathcal{D}_u|)\big).$$

The first term corresponds to weak-label generation and hidden-state extraction, while the second term accounts for training the NTF for $e$ epochs on $\mathcal{D}_\ell$ and scoring $\mathcal{D}_u$. This procedure is cost-efficient because the expensive operation, the weak-teacher forward pass, is already required for weak-label generation, and the same pass also provides the hidden states used by the NTF. The NTF itself is only a small MLP applied to cached hidden states, so its training and scoring cost is negligible when

$$C_{\text{NTF}}(e \cdot |\mathcal{D}_\ell| + |\mathcal{D}_u|) \ll \bar{C}_{\text{teacher}}(|\mathcal{D}_\ell| + |\mathcal{D}_u|).$$

Thus, in our setting, the pipeline is dominated by teacher inference rather than by fitting or applying the trust function.

**Training and Evaluation.** To train $\tau$, we construct binary supervision from a labeled source dataset by marking each reference prediction as correct or incorrect under the task-specific evaluation rule (exact match for MCQA and mathematical problems, and best-move match for chess puzzles). We fit $\tau$ to predict this correctness indicator from $g_{\pi_\mathcal{W}}(x, \hat{y})$ using the optimization described earlier. At evaluation time, we apply the trust function to held-out data and report standard reliability metrics such as AUC (Hanley & McNeil, 1982), ECE (Guo et al., 2017), and Brier score (Brier, 1950).

We additionally report *purity*, which is the fraction of correct examples among those retained by selecting the top-trust subset used for downstream training. Table 1 reports these metrics for neural trust functions across domains.

| Domain | AUC (↑) | ECE (↓) | Brier (↓) | Purity (↑) |
|---|---|---|---|---|
| World Knowledge | 0.92 | 0.03 | 0.07 | 0.98 |
| Quantitative Reasoning (OMNI) | 0.83 | 0.11 | 0.13 | 0.69 |
| Quantitative Reasoning (MATH) | 0.84 | 0.14 | 0.17 | 0.95 |
| Strategy Games | 0.91 | 0.02 | 0.11 | 0.95 |

*Table 1.* **NTF evaluation results (§2.2).** World knowledge and strategy games use Qwen3-0.6B as the teacher. Quantitative reasoning uses Qwen3-1.7B (OMNI-MATH) and Gemma3-1B (MATH). See App. A.6 for full results.

**Justification for Labeled Data Assumption.** Neural trust functions require labeled data, but crucially *not* from the target distribution where labels are unavailable. This assumption is reasonable for two distinct reasons. First, supervision is often *unevenly distributed*. Labels are abundant for well-studied benchmarks, while scarcity arises in underrepresented targets. We exploit this by training $\tau$ on an abundant labeled source distribution and deploying it zero-shot under in-domain distribution shift (§3). For example, in quantitative reasoning, we evaluate students on AIME, where labeled training data is limited, while training $\tau$ on easier and widely available supervision such as MATH. Second, we empirically find that $\tau$ remains useful even when supervision must come from an extremely weak teacher. For example, Qwen3-1.7B achieves $< 5\%$ accuracy on AIME, yet trust-filtered weak supervision still achieves near-lossless recovery relative to ground-truth training (§4).

## 3. Weak-to-Strong Generalization by Learning to Trust

### 3.1. Learning to Trust Framework

**Prerequisites.** We consider two models, a *weak* teacher $\pi_\mathcal{W}$ and a *strong* student $\pi_\mathcal{S}$. We define *weak* and *strong* models by their *initial capacity*, specified by model size (Medvedev et al., 2025). We require a labeled source dataset $\mathcal{D}_\ell = \{(x_i, y_i)\}_{i=1}^{N_\ell}$ and an unlabeled target dataset $\mathcal{D}_u = \{x_j\}_{j=1}^{N_u}$, where $\mathcal{D}_\ell$ and $\mathcal{D}_u$ are not necessarily drawn from the same distribution. Our goal is to improve $\pi_\mathcal{S}$ on $\mathcal{D}_u$ without access to its ground-truth labels.

The top-left panel of Fig. 1 summarizes the *Learning to Trust* (L2T) framework, which serves as the shared protocol for all experiments reported in §4. A weak teacher $\pi_\mathcal{W}$ produces weak labels $\hat{y}$ on inputs $x \in \mathcal{D}_u$. To estimate which weak labels are reliable enough to use as supervision, we train a neural trust function $\tau$ on $\mathcal{D}_\ell$ (§2) to predict weak label correctness from the teacher's internal representation $g_{\pi_\mathcal{W}}(x, \hat{y})$. At deployment, we score each weak label by $t =$

| Teacher → | **OLMo2-1B** | | | **Qwen3-0.6B** | | | | |
|---|---|---|---|---|---|---|---|---|
| Method ↓ – Student → | OLMo2-1B | OLMo2-7B | OLMo2-13B | Qwen3-0.6B | Qwen3-1.7B | Qwen3-4B | Qwen3-8B | Qwen3-14B |
| Teacher Performance | **43.1** | | | **52.6** | | | | |
| No-SFT | 43.1 | 65.1 | 73.8 | 52.6 | 69.1 | 72.0 | 77.3 | 79.4 |
| Naive | 49.0 (90.8) | 69.3 (48.3) | 74.7 (12.2) | 59.2 (71.0) | 74.0 (86.0) | 78.5 (75.6) | 83.8 (82.3) | 86.0 (86.8) |
| I-Confidence | 49.2 (93.8) | 69.2 (47.1) | 75.1 (17.6) | 59.4 (73.1) | 74.3 (91.2) | 78.8 (79.1) | 83.8 (82.3) | 85.7 (82.9) |
| ICL + I-Confidence | 50.0 (106.2) | 72.0 (79.3) | 77.9 (55.4) | **61.8** (98.9) | 74.4 (93.0) | 79.4 (86.0) | 84.8 (94.9) | 86.5 (93.4) |
| Ensemble | 49.2 (93.8) | 70.9 (66.7) | 76.7 (39.2) | 59.7 (76.3) | 74.1 (87.7) | 77.2 (60.5) | 82.9 (70.9) | 85.1 (75.0) |
| Reward Model | 46.1 (46.2) | 68.8 (42.5) | 78.4 (62.2) | 59.0 (68.8) | 71.7 (45.6) | 77.0 (58.1) | 82.6 (67.1) | 86.1 (88.2) |
| **NTF (Ours)** | 50.5 (113.9) | 73.7 (98.9) | 80.9 (95.9) | 61.6 (96.8) | **75.0** (103.5) | 80.0 (93.0) | **84.9** (96.2) | **87.1** (101.3) |
| Ground Truth | 49.6 | 73.8 | 81.2 | 61.9 | 74.8 | 80.6 | 85.2 | 87.0 |

*Table 2.* **Average world knowledge results across benchmarks (§4.1).** Accuracy (%) averaged over OBQA, ARC-C, ARC-E, SCIQ, and SIQA. Each column corresponds to a student trained under the teacher shown above. Recovery (Eq.1) for each baseline is reported inside the parentheses. For **NTF**, cell colors encode the paired significance test between **NTF** and **GT**: dark green means NTF is *significantly* better than GT (super-recovery), light green means *no significant difference* (near-lossless recovery), and light red means GT is *significantly* better than NTF. Differences are assessed with an exact paired test ($\alpha = 0.05$). See App. G for significance test details.

$\tau(g_{\pi_W}(x, \hat{y}))$ and *filter* the weakly labeled pool by retaining the highest-trust examples, yielding a high-trust training set $\tilde{\mathcal{D}}_u$. We then train a strong student $\pi_S$ on $\tilde{\mathcal{D}}_u$ using a fixed downstream training recipe (e.g., SFT or GRPO).

### 3.2. Experimental Setup

**Domains.** We evaluate weak-to-strong generalization across three *domains*: (i) world knowledge, (ii) quantitative reasoning, and (iii) strategy games. Each domain is instantiated via a fixed *task interface* (input-output format): world knowledge uses multiple-choice question answering (MCQA), quantitative reasoning uses mathematical problem solving, and strategy games uses chess puzzles. Throughout, we refer to these settings by *domain names* and use the interface terms only when discussing benchmarks or evaluation.

**Models.** We study two LLM families and a range of scales. For world knowledge and strategy games, we use the OLMO2 family (1B–13B) (OLMo et al., 2025) and the QWEN3 family (0.6B–14B) (Yang et al., 2025), with the smallest model in each family serving as the default weak teacher. For quantitative reasoning, we primarily use QWEN3 teachers and students (1.7B–8B), and additionally include Gemma3-1B (Team et al., 2025) as a teacher when training a LLaMA3.1-8B (Grattafiori et al., 2024) student. All results are reported for pretrained checkpoints, unless noted (App. D). We list all model used in App. J.

**Generalization Regimes.** We evaluate neural trust functions under three regimes, distinguished by the relationship between the labeled source distribution underlying $\mathcal{D}_\ell$, on which $\tau$ is trained, and the target distribution underlying $\mathcal{D}_u$, on which $\tau$ is applied. **ID** (in-distribution) evaluates $\tau$ on held-out examples from the same data distribution (benchmark) used to train $\tau$. **OOD$_{dist}$** (in-domain distribution shift) trains $\tau$ on one benchmark and applies it zero-shot to a different benchmark with the *same task interface* (i.e., the same

output space) but a different data distribution (e.g., MMLU → ARC-EASY for MCQA). **OOD$_{domain}$** (out-of-domain) applies $\tau$ across a *changed task interface* (e.g., MCQA → chess puzzles), which is a strictly stronger shift than OOD$_{dist}$. Unless stated otherwise, our claims about zero-shot transfer refer to OOD$_{dist}$. In Table 1, we report ID performance for strategy games and OOD$_{dist}$ performance for world knowledge and quantitative reasoning. Additional regime-specific evaluations are provided in App. B. Across our experiments, neural trust functions transfer reliably in ID and OOD$_{dist}$ settings, but degrade when applied OOD$_{domain}$, suggesting that trust is partly tied to the task interface.

**Baselines.** We compare NTF against a set of weak-to-strong supervision baselines that differ only in how they select examples from the weakly labeled pool. *No-SFT* and *No-GRPO* evaluate the student without additional training. *Naive* trains on weak labels from randomly selected examples, using difficulty-stratified sampling for quantitative reasoning and uniform sampling for other domains. *Internal Confidence* selects examples on which the weak teacher assigns high length-normalized log-probability to its predicted label, while *ICL + Internal Confidence* computes the same score after prepending five fixed in-context demonstrations. *Verbalized Confidence* instead asks the weak teacher to report its own confidence and selects examples with the highest self-reported scores. *Ensemble* selects examples on which two independent weak teachers agree, measuring the value of multi-teacher consistency. *Reward Model* baselines select examples using external public reward models or verifiers. Finally, *Ground Truth* trains the student on budget-matched gold labels from the target domain, providing an oracle supervised reference. We provide a detailed description of each baseline in App. F.

**Evaluation.** We evaluate the resulting student on held-out labeled data from the target domain using the domain-

standard metric (e.g., accuracy). Evaluation details are provided in App. E. When comparing to ground-truth training, we summarize how much ground-truth supervised improvement is recovered by each baseline via

$$\text{Recovery} = \frac{\text{Baseline} - \text{Base}}{\text{GT} - \text{Base}} \times 100\%. \quad (1)$$

where Base denotes the no-training baseline for the domain (e.g., No-SFT or No-GRPO), and GT denotes training on $n$ ground-truth labeled target examples. We refer to *near-lossless recovery* as settings where NTF and GT are not statistically distinguishable, regardless of which one is numerically higher and to *super-recovery* as settings where NTF is statistically better than GT (App. G). For strategy games, the no-training baseline is 0.0% for all models under our evaluation setup; therefore, we omit the No-SFT row from tables for brevity (see App. D.3 for details).

## 4. Results

### 4.1. World Knowledge

**Setup.** Under $\text{OOD}_{\text{dist}}$, we train $\tau$ on labeled MMLU (Hendrycks et al., 2021a) and use it to score weak labels on training splits of target MCQA benchmarks (OPENBOOKQA (Mihaylov et al., 2018), ARC-CHALLENGE/EASY (Clark et al., 2018), SCIQ (Welbl et al., 2017), SOCIALIQA (Sap et al., 2019)). We then LoRA-SFT (Hu et al., 2021) students using the top-$n$ examples by trust (App. D.1) and evaluate on associated test splits (App. E.1).

**Results.** Table 2 shows that *trust filtering consistently improves over unfiltered weak supervision* (Naive), confidence-based heuristics (I-Confidence, ICL + I-Confidence), multi-teacher supervision (Ensemble), and off-the-shelf reward-model filtering (Reward Model). Across model families and scales, NTF closely matches Ground Truth, with **NTF being statistically indistinguishable from Ground Truth in 5 of the 8 settings and significantly better in 1 setting** (App. G). Compared to reward models, NTF is consistently stronger, suggesting that teacher hidden representations provide a more reliable signal for weak label correctness than generic output-level reward scoring in the MCQA weak-to-strong setting. This supports our claim that *Learning to Trust* achieves near-lossless recovery and occasionally super-recovery using only teacher-generated weak labels. We provide per-benchmark results in App. K.

### 4.2. Quantitative Reasoning

**Setup.** Under $\text{OOD}_{\text{dist}}$, we train $\tau$ on labeled MATH (Hendrycks et al., 2021b) and apply it to teacher rollouts on OMNI-MATH (Gao et al., 2024). We then train the student on the top-$n$ rollouts ranked by trust scores using GRPO (Shao et al., 2024) (App. D.2), and evaluate the student on AIME (Veeraboina, 2023) (App. E.2).

| Teacher → | Qwen3-1.7B | | Qwen3-4B | Qwen3-8B |
|---|---|---|---|---|
| Method ↓ – Student → | Qwen3-4B | Qwen3-8B | Qwen3-8B | Qwen3-8B |
| Teacher Performance | **4.27** | | **10.91** | **18.88** |
| No-GRPO | 10.9 | 18.9 | 18.9 | 18.9 |
| Naive | 19.6 (72.5) | 26.0 (83.5) | 27.2 (84.7) | 26.8 (83.2) |
| I-Confidence | 20.7 (81.7) | 25.6 (78.8) | 26.7 (79.6) | 27.3 (88.4) |
| V-Confidence | 18.7 (65.0) | 25.9 (82.4) | 27.1 (83.7) | 26.5 (80.0) |
| **NTF (Ours)** | **22.0** (92.5) | **26.6** (91.0) | **27.9** (91.7) | **27.4** (89.2) |
| Ground Truth | 22.9 | 27.4 | 28.7 | 28.4 |

*Table 3.* **Quantitative reasoning results (§4.2).** Each column is a student trained under the teacher shown above. Recovery (Eq.1) for each baseline is reported inside the parentheses. Significance tests follow from Table 2.

**Results.** Table 3 shows that trust filtering is an effective replacement for ground-truth supervision. Across QWEN3 teacher-student pairs, NTF consistently outperforms unfiltered weak supervision (Naive) and confidence-based selection (I-Confidence, V-Confidence), and typically approaches Ground Truth. The advantage over confidence-based selection is largest for weaker teachers (e.g., Qwen3-1.7B), suggesting that representation-level trust better separates correct from incorrect rollouts when likelihood-based confidence is miscalibrated on reasoning traces. Recovery is consistently high (89–92%), and **in half of the settings NTF is statistically indistinguishable from Ground Truth**, supporting our near-lossless recovery characterization (App. G). We address concerns about spurious rewards (Shao et al., 2025) in App. C and report additional baselines in App. F.

### 4.3. Strategy Games

**Setup.** Under ID, we train $\tau$ on a labeled subset of LICHESS puzzles (Lichess, n.d.) and apply it to score weak labels on a disjoint subset. We then LoRA-SFT (Hu et al., 2021) students using the top-$n$ puzzles by trust (App. D.3) and evaluate on held-out test puzzles (App. E.3).

**Results.** Table 4 shows that NTF consistently outperforms unfiltered weak supervision (Naive) and confidence-based filtering (I-Confidence, V-Confidence) across both teacher families and nearly all student scales. The gains are most pronounced in the QWEN3 family, where trust filtering closes essentially all of the gap to Ground Truth and often surpasses it. This is reflected in the significance tests, where **NTF exhibits super-recovery in 4 of the 8 settings and is near-lossless in 1 setting** (App. G). In contrast, the OLMO2 family is less reliable in this domain, so the weak-to-strong gap is harder to eliminate, though NTF remains the strongest weak-supervision strategy and consistently narrows the gap to Ground Truth.

## 5. Snowballing Weak-to-Strong Generalization

We now ask whether trust filtering supports not only one-shot weak-to-strong transfer, but also an *iterative* training

| Teacher → | **OLMo2-1B** | | | **Qwen3-0.6B** | | | | |
|---|---|---|---|---|---|---|---|---|
| Method ↓ – Student → | OLMo2-1B | OLMo2-7B | OLMo2-13B | Qwen3-0.6B | Qwen3-1.7B | Qwen3-4B | Qwen3-8B | Qwen3-14B |
| Teacher Performance | **28.5** | | | **6.9** | | | | |
| Naive | 34.7 (92.0) | 37.0 (69.9) | 38.3 (70.3) | 10.6 (71.1) | 22.1 (96.1) | 27.0 (74.4) | 33.7 (91.1) | 38.1 (95.5) |
| I-Confidence | 37.1 (98.4) | 39.9 (75.4) | 40.7 (74.7) | 11.5 (77.2) | 23.0 (100.0) | **36.9 (101.7)** | 36.3 (98.1) | 37.5 (94.0) |
| V-Confidence | 35.2 (93.4) | 37.7 (71.3) | 38.7 (71.0) | 7.3 (49.0) | 17.9 (77.8) | 31.8 (87.6) | 31.8 (85.9) | 33.4 (83.7) |
| **NTF (Ours)** | **37.4 (99.2)** | **41.2 (77.9)** | **41.5 (76.1)** | **15.5 (104.4)** | **25.4 (110.5)** | **35.4 (97.6)** | **38.0 (102.9)** | **44.1 (110.4)** |
| Ground Truth | 37.7 | 52.9 | 54.5 | 14.9 | 23.0 | 36.3 | 37.0 | 39.9 |

*Table 4.* **Strategy games results (§4.3).** Each column is a student trained under the teacher shown above. Recovery (Eq.1) for each baseline is reported inside the parentheses. Significance tests follow from Table 2.

mechanism that compounds gains across generations. We study this question in the strategy games domain, where the large-scale LICHESS dataset provides enough training examples to train neural trust functions and successive students over multiple iterations. Starting from the weakest available teacher (Qwen3-0.6B), we repeatedly apply trust-filtered supervision, using the student from iteration $k-1$ as the teacher at iteration $k$, which yields a *weak-to-strong chain* that culminates in Qwen3-14B. This design targets settings where teacher competence is uncertain a priori and is motivated by our earlier results, where training on trust-filtered weak labels often matches or exceeds ground-truth training, suggesting a *snowballing effect*.

| Method | Qwen3-4B | Qwen3-8B | Qwen3-14B |
|---|---|---|---|
| Naive Shallow (0.6B) | 27.0 | 33.7 | 38.1 |
| Naive Chain | 30.1 | 34.2 | 39.1 |
| NTF Shallow (0.6B) | 35.4 | 38.0 | 44.1 |
| NTF Shallow (8B) | — | — | 46.1 |
| **NTF Chain** | **36.9** | **40.1** | **48.2** |
| Ground Truth | 36.2 | 37.0 | 40.0 |

*Table 5.* **Weak-to-strong chaining results on the strategy games domain.** Compared to shallow one-step transfer, NTF chaining yields compounding gains across student scales and outperforms ground-truth supervision at the largest scale.

As shown in Table 5, iterative weak-to-strong training yields *compounding* improvements throughout the chain. The final chained Qwen3-14B outperforms (i) a single-step transfer trained directly from the weakest teacher (Qwen3-0.6B), (ii) a single-step transfer trained from the strongest available weak teacher (Qwen3-8B), (iii) the corresponding naive chaining baseline, and (iv) budget-matched training with ground-truth supervision. The intermediate students show the same pattern, where NTF chaining consistently improves over shallow NTF transfer and naive supervision, with the advantage widening across iterations. Together, these results show that chaining amplifies the gains from trust-filtered weak supervision, yielding increasing returns across iterations.

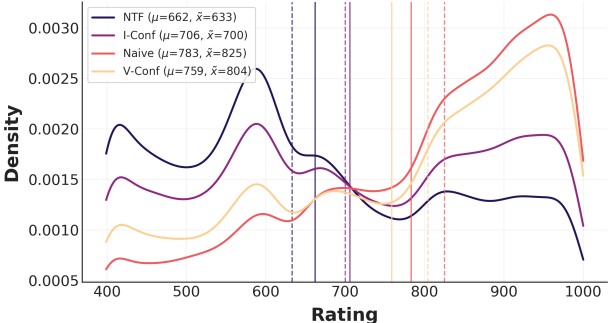

*Figure 2.* **Rating distributions of selected training examples under Qwen3-0.6B (§6.1).** Compared to Naive and confidence-based selection, NTF shifts mass toward lower-rated puzzles.

# 6. Mechanisms Behind Near-Lossless Weak-to-Strong Generalization

In several settings, students trained on trust-filtered weak labels match or outperform students trained on ground-truth labels, despite the filtered weak labels being high-purity but imperfect (App. A.6). To understand why, we analyze how different selection rules change the *composition* of the resulting training set and how these shifts shape learning. We conduct our analysis in the strategy games domain (§4.3) to avoid data contamination concerns that can complicate interpretation (Kocyigit & Yildirim, 2026).

## 6.1. Finding 1: Neural Trust Functions are Conservative

We first test whether trust filtering changes the difficulty profile of the selected training set. We analyze this effect in the strategy games domain, as LICHESS puzzles come with an explicit, fine-grained rating that serves as a ground-truth difficulty signal. Fig. 2 compares the rating distribution of chess puzzles selected by each filter. When applied to Qwen3-0.6B, NTF is noticeably conservative in difficulty. It concentrates selection on lower-rated puzzles, reducing both the mean and median rating relative to Naive and confidence-based filters. Since Naive samples uniformly from the data distribution, it closely matches the underlying

rating distribution. This shift suggests that NTF reweights the training set toward easier instances, which act like an implicit easy-first curriculum in addition to filtering label noise. We show a similar data distribution in quantitative reasoning data selection and the result in App. H.

To quantify how much of NTF's improvement is explained by this curriculum shift, we construct a difficulty-matched baseline (Naive-DM). We bin puzzles by rating, match the per-bin proportions of the NTF-retained subset, then sample puzzles uniformly from each bin. This preserves NTF's difficulty profile while keeping the label purity comparable to Naive, up to sampling noise. Table 6 shows that difficulty matching helps for smaller students (1.7B, 4B), recovering part of the gap between Naive and NTF. However, the effect does not persist at larger scales (8B, 14B), where Naive-DM is similar to or slightly worse than Naive. These results suggest that difficulty reweighting may be one contributor to NTF's gains at smaller students, but it is unable to explain the improvements observed at larger scales.

| Method | Qwen3-1.7B | Qwen3-4B | Qwen3-8B | Qwen3-14B |
|---|---|---|---|---|
| NTF (Ours) | 25.4 | 35.4 | 38.0 | 44.1 |
| Naive | 22.1 | 27.0 | 33.7 | 38.1 |
| Naive-DM | 24.8 | 29.7 | 32.7 | 35.9 |

*Table 6.* **Difficulty-and-purity-matched ablation on strategy games domain (§6.1).** Naive-DM matches the rating distribution of the NTF while keeping label purity comparable to Naive, isolating gains due to difficulty reweighting.

## 6.2. Finding 2: Neural Trust Functions Often Recover Optimal Alternatives

We then analyze examples where the trust function assigns high trust to a teacher's move that the ground truth marks as incorrect. We study this in the strategy games domain because, unlike world knowledge and quantitative reasoning, it admits an automated evaluator for the plausibility of alternative labels. Concretely, we use Stockfish (The Stockfish Developers, 2025) to compare the NTF-retained move with the ground-truth move. We compute the *advantage gap* of the two moves as $\Delta = \text{Advantage}_{\text{GT}} - \text{Advantage}_{\text{NTF}}$, where $\text{Advantage}_{\text{GT}}$ denotes the Stockfish-evaluated advantage after the ground-truth move and $\text{Advantage}_{\text{NTF}}$ denotes the advantage after the move retained by NTF. A larger gap indicates that the NTF-retained move is worse relative to the ground-truth move under Stockfish's evaluation. This helps distinguish trust-function failures from the suboptimality of ground-truth label. In this context, we define *optimal alternatives* as instances where the NTF-retained move yields a larger engine advantage (higher Stockfish score) than the ground-truth move.

Fig. 3 plots the distribution of these gaps. The distribution places substantial mass on negative values, which indicates

that the NTF-retained moves are often stronger than the ground truth best move under engine evaluation. Furthermore, 66.1% of NTF-retained moves lead to a winning mate. Together, these patterns suggest that many apparent false positives arise from the suboptimality of ground truth label rather than trust-function error.

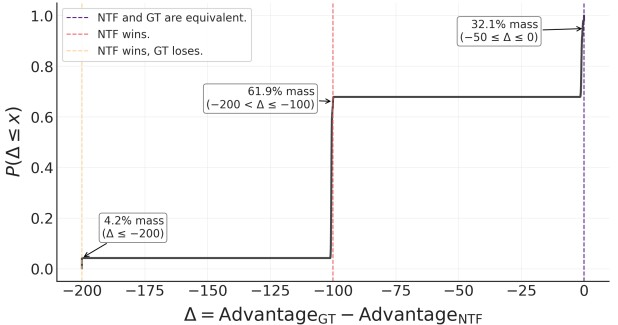

*Figure 3.* **Stockfish advantage gaps for NTF *false positives* (§6.2).** Distribution of Stockfish-evaluated advantage gaps between the GT best move and the NTF-retained move, restricted to examples where the dataset marks the NTF move as incorrect. Values near zero indicate near-equivalent moves, while the negative values indicate the NTF move is stronger.

We then test whether these label-incomplete cases contribute positively to learning by fixing the same NTF-retained inputs and replacing the teacher moves with ground-truth moves (NTF-GT). Since NTF selection has 95.27% purity, this replacement changes the training label on only 4.73% of examples. Table 7 shows that NTF-GT consistently underperforms NTF by a small margin across scales. Since only 4.73% of labels are changed, this drop implies that some moves labeled incorrect by the dataset are nevertheless strong alternatives that provide useful supervision. At the same time, NTF-GT remains competitive with, and often exceeds, standard ground-truth training (GT), which indicates that the main gains come from data selection. This also suggests that recovering improved labels on the small set of label-ambiguous examples contributes to performance, but it is not the primary driver of NTF's advantage.

| Method | Qwen3-1.7B | Qwen3-4B | Qwen3-8B | Qwen3-14B |
|---|---|---|---|---|
| Ground Truth | 23.0 | 36.3 | 37.0 | 39.9 |
| NTF (Ours) | 25.4 | 35.4 | 38.0 | 44.1 |
| NTF-GT | 24.8 | 35.3 | 36.2 | 43.6 |

*Table 7.* **Ground-truth relabeling on the NTF-retained subset on strategy games domain (§6.2).** NTF-GT keeps the same NTF-retained examples but replaces teacher moves with GT labels.

## 6.3. Finding 3: Neural Trust Functions Induce More Coherent Gradients

We finally analyze how trust filtering changes the learning signal in strategy games domain. For each selection rule, we

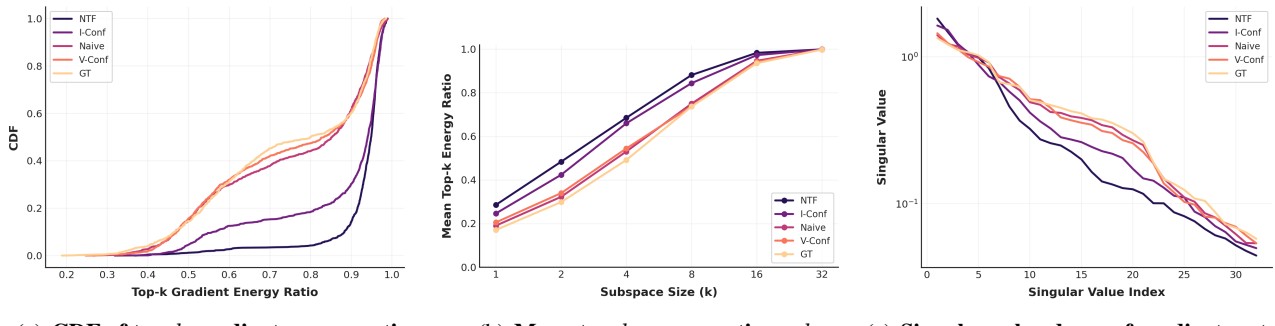

(a) **CDF of top-$k$ gradient energy ratio**    (b) **Mean top-$k$ energy ratio vs. $k$**    (c) **Singular value decay of gradient matrix**

*Figure 4.* **Gradient-subspace alignment diagnostics on strategy games domain (§6.3). (Left)** Empirical CDF of per-example *top-k gradient energy ratio* ($k$=8). **(Middle)** Mean top-$k$ energy ratio as a function of $k$ ($k \in \{1, 2, 4, 8, 16, 32\}$). **(Right)** Top-32 singular values of the gradient matrix. Across panels, NTF concentrates more gradient energy in a low-dimensional subspace and exhibits faster singular-value decay, indicating more coherent (lower-rank) update directions.

compute per-example gradients of the label loss with respect to the student's last-layer hidden states. We then measure how strongly these gradients concentrate in a shared low-dimensional subspace given by the top-$k$ singular directions of the gradient matrix.

Fig. 4 presents three complementary diagnostics. First, the CDF of the top-$k$ gradient energy ratio (left, $k$=8) shows that NTF-retained examples consistently concentrate more gradient energy in the dominant subspace than all other baselines, indicating a distributional shift toward more aligned update directions. Second, sweeping $k$ (middle) reveals that this advantage persists across small to moderate subspace dimensions and only saturates when $k$ becomes large, where all methods necessarily approach full energy coverage. Finally, the singular value spectrum of the per-method gradient matrix (right) exhibits a steeper decay under NTF, suggesting a lower-rank and coherent gradient structure.

Overall, NTF selects data that yields more consistent gradient directions across examples. This reduces gradient diversity without removing the training signal, which helps explain the improved stability and sample efficiency we observe with trust filtering.

## 7. Risk-Controlled Data Selection

Our main experiments use a fixed-budget protocol, where all baselines retain the same number of weakly labeled examples to isolate the effect of selection quality. In practice, however, the appropriate retention budget or score threshold may be unknown. We therefore provide a risk-controlled calibration procedure that uses a small labeled target calibration set to choose a trust-score threshold using a finite-sample upper confidence bound on selected-label noise.

For a candidate threshold $\theta$, let $\widehat{r}(\theta)$ denote the empirical noise rate among calibration examples with trust score at

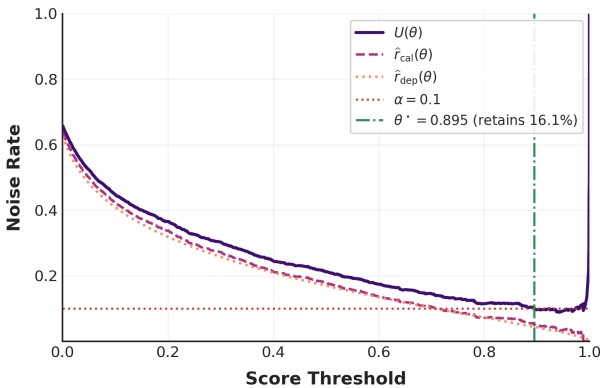

*Figure 5.* **Risk-controlled threshold calibration.** $\widehat{r}_{\mathrm{cal}}(\theta)$ is the empirical noise rate at threshold $\theta$ on a small labeled calibration subset, $U(\theta)$ is its Hoeffding upper confidence bound, and $\widehat{r}_{\mathrm{dep}}(\theta)$ is the noise rate on the held-out deployment pool. We select the most inclusive $\theta^\star$ satisfying $U(\theta) \leq \alpha = 0.1$, giving $\theta^\star = 0.895$ which retains 16.1% of the deployment pool; the realized $\widehat{r}_{\mathrm{dep}}(\theta^\star)$ stays below $\alpha$.

least $\theta$, and let $n(\theta)$ denote the number of selected calibration examples. For any fixed threshold with $n(\theta) > 0$, Hoeffding's inequality (Hoeffding, 1963) gives the upper confidence bound $U(\theta) = \widehat{r}(\theta) + \sqrt{\frac{\log(1/\delta)}{2n(\theta)}}$. That is, the true selected-label noise rate at this fixed threshold is at most $U(\theta)$ with probability at least $1 - \delta$ under the calibration distribution. Given a target noise rate $\alpha$ and confidence parameter $\delta$, we choose the most inclusive threshold satisfying $U(\theta) \leq \alpha$. In this section, we use this uncorrected per-threshold bound as a practical calibration rule, and defer the effect of multiple-testing correction across threshold candidates to App. I.

Fig. 5 shows that risk-controlled calibration selects a high-purity subset without manually tuning a threshold. This result demonstrates that trust scores computed by NTFs are

not only discriminative but also sufficiently aligned with label correctness to support calibrated selection. In particular, higher trust scores correspond to cleaner weak labels, allowing the finite-sample UCB to produce an operationally useful threshold rather than a vacuously conservative one. Empirically, the selected threshold retains a large number of weakly labeled examples while the measured deployment noise rate remains below the target level, providing a practical alternative to fixed-budget selection when the appropriate retention budget is unknown.

## 8. Related Work

**Limitations of Weak Supervision.** Principles of weak-to-strong generalization have been explored in several domains, including but not limited to easy-to-hard generalization (Sun et al., 2024), preference learning (Tao & Li, 2025), reward modeling (Hauptvogel et al., 2024), and planning (Ye et al., 2026; Ding et al., 2026). While early weak-to-strong generalization research demonstrates that student models can surpass their weak teachers, a persistent performance gap remains between weakly supervised students and models trained on ground-truth labels (Burns et al., 2023; Agrawal et al., 2024; Lang et al., 2025). Recent theory attributes this ceiling to error propagation, where students inherit systematic teacher mistakes unless the data geometry explicitly enables correction (Lang et al., 2024; Wu & Sahai, 2024), as well as representation gaps, where weak labels fail to span task-relevant directions that lie outside the teacher's representational capacity (Xue et al., 2025). Furthermore, weak-to-strong gains are characterized in terms of a *misfit* between the student model class and the weak supervision signal, suggesting that reducing harmful supervision can improve generalization (Charikar et al., 2024). Empirically, these errors are catastrophically amplified under distribution shift, where confidently wrong weak labels can destabilize student learning entirely (Yu et al., 2021). These limitations motivate the need for mechanisms that selectively trust weak supervision rather than treating all weak labels equally.

**Trust Estimation.** When weak-to-strong generalization is framed as a data selection problem, the key challenge is to identify which weak labels are correct. Data selection and reweighting have also been studied in the weak supervision literature (Dehghani et al., 2017; Lang et al., 2022), but these works primarily target in-domain weak supervision rather than weak-to-strong generalization under distribution shift. Most prior work relies on *output-level* confidence proxies such as predictive entropy (Guo & Yang, 2024), self-consistency (Wang et al., 2023), or inter-model agreement (Lang et al., 2025). Learned verifiers and reward models are also related, as they score whether a model output is reliable (Cobbe et al., 2021); however, they typically operate on textual input-output pairs, while our trust functions use the weak teacher's hidden representations. Yet, these output-level signals are often miscalibrated on complex tasks, assigning high confidence to fluent but incorrect answers (Tian et al., 2023; Fadeeva et al., 2023). Recent evidence suggests that correctness is reflected in *internal* model representations, even when a model's final output is wrong (Azaria & Mitchell, 2023; Zhang et al., 2025). Motivated by this, we learn neural trust functions over internal activations and use them to verify weak labels, aiming for more reliable selection than output-based heuristics.

**Training Frameworks for Weak-to-Strong Generalization.** Beyond data selection, several works propose modifying the training procedure itself to mitigate weak supervision noise. These include multi-stage pipelines combining filtering with preference optimization (Yang et al., 2024b; Somerstep et al., 2024), ensemble-based supervision (Agrawal et al., 2024), and iterative weak-to-strong chains (Liang et al., 2024). In parallel, several work suggests that reasoning can be improved with weak supervision by bootstrapping from a small labeled seed (Tong et al., 2024), and that suitably designed weak signals can incentivize stronger reasoning without expensive demonstrations (Yuan et al., 2026). While effective, these methods often require multiple teachers, repeated querying, or complex multi-step pipelines. In contrast, our approach is orthogonal to training-time modifications, as it focuses exclusively on improving supervision quality prior to training through efficient trust estimation.

## 9. Conclusion

We study weak-to-strong generalization as a data selection problem. We formalize this view with *trust functions* and introduce *neural trust functions* that predict weak label correctness from teacher internal representations. Across multiple domains, trust-filtered weak supervision often matches training on ground-truth labels and sometimes exceeds it, yielding near-lossless weak-to-strong generalization. We further show that trust filtering enables a weak-to-strong chain where each trained student becomes the next teacher, which produces compounding improvements across iterations. Our analyses indicate that NTFs help beyond reducing label errors, as they bias selection toward easier instances, surface strong alternatives when ground truth is incomplete, and produce more coherent gradient signals.

Future work could extend neural trust functions beyond single-token hidden states to aggregate evidence across tokens, layers, or reasoning steps; evaluate trust filtering under more realistic weak-supervision sources such as synthetic data, noisy human labels, retrieval-augmented teachers, and multi-teacher pipelines.

## Impact Statement

This paper presents a method for improving the data efficiency of large language model training. By enabling high-performance student models to learn from weaker, abundant, or synthetic supervision, our approach reduces the barrier to entry for training capable models in domains where ground-truth data is scarce or costly to collect. This has potential positive impacts for democratizing access to high-quality models in specialized domains (e.g., science, medicine). However, we note that relying on automated data selection carries the risk of amplifying biases present in the weak teacher's internal representations. Future work should investigate whether trust filtering disproportionately selects or suppresses specific demographic or ideological viewpoints.

## Limitations

While our framework yields near-lossless weak-to-strong generalization, we identify a few boundaries of our current approach that warrant further investigation. First, unlike unsupervised heuristics, neural trust functions require a labeled source dataset to learn the mapping from representations to correctness, limiting their applicability in regimes with absolutely no ground truth. Second, we focus solely on outcome supervision (i.e., predicting the correctness of the final answer), leaving the potential benefits of dense, step-wise supervision (e.g., process rewards) for complex reasoning tasks unexplored. Lastly, we restrict our architectural scope to simple MLPs acting on single-token hidden states, potentially missing temporal reasoning signals that richer architectures such as attention-based NTFs might capture.

## Acknowledgments

We acknowledge the use of computational resources on the Johns Hopkins Data Science and AI Institute (DSAI) cluster. We sincerely thank Jack Zhang and Tianjian Li for their insightful discussions and support for our work, as well as the JHU CLSP and DSAI communities for their helpful comments and feedback.

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

| Layer $\ell$ | Brier ($\downarrow$) | AUC ($\uparrow$) | ECE ($\downarrow$) |
|---|---|---|---|
| 1 | 0.1853 | 0.7650 | 0.0450 |
| 5 | 0.1298 | 0.8862 | 0.0364 |
| 9 | 0.1147 | 0.9155 | 0.0412 |
| 13 | 0.1033 | 0.9284 | 0.0350 |
| 17 | 0.0977 | 0.9341 | 0.0270 |
| 21 | 0.0906 | 0.9426 | 0.0198 |
| 25 | 0.0808 | 0.9548 | **0.0139** |
| 28 | **0.0801** | **0.9562** | 0.0197 |

*Table 8.* Layer ablation for neural trust functions on chess using `Qwen3-0.6B`.

| Token Position | AUC ($\uparrow$) | ECE ($\downarrow$) | Brier ($\downarrow$) |
|---|---|---|---|
| Last Input Token | 0.7435 | 0.0518 | 0.1837 |
| Last Output Token | **0.9562** | **0.0197** | **0.0801** |

*Table 9.* Token-position ablation for neural trust functions on chess using `Qwen3-0.6B`.

| Pooling | AUC ($\uparrow$) | ECE ($\downarrow$) | Brier ($\downarrow$) |
|---|---|---|---|
| Mean Pooling | 0.7704 | 0.0457 | 0.1726 |
| No Pooling | **0.9562** | **0.0197** | **0.0801** |

*Table 10.* Pooling ablation for neural trust functions on chess using `Qwen3-0.6B`.

## A. Neural Trust Function Ablations

### A.1. Input Ablations

We ablate design choices for the *input representation* used by neural trust functions. We train a 4-layer MLP trust function with hidden width 512 on 50,000 chess training examples and evaluate on a disjoint set of 1,000 chess examples, using `Qwen3-0.6B` as the weak teacher. Across ablations, we keep the NTF's architecture, optimization setup, and training budget fixed, and vary only how the teacher representation is constructed (layer choice, token position, and pooling). We note that these ablations are conducted on a separate data split from the datasets used in the main experiments.

**Layer choice.** We vary which transformer layer provides the hidden representation to the trust function. For each layer $\ell$, we extract the hidden state at layer $\ell$ for the final generated token and feed it directly to the NTF (no pooling), isolating the effect of layer depth on correctness separability. Specifically, for `Qwen3-0.6B` (28 layers), we evaluate layers at a stride of four, extracting representations from $\ell \in \{1, 5, 9, 13, 17, 21, 25, 28\}$. Table 8 shows that performance improves with depth. Late layers achieve the best accuracy (Brier) and discrimination (AUC) (best at $\ell=28$), while calibration (ECE) peaks slightly earlier (best at $\ell=25$). We therefore use last-layer representations by default.

**Token position.** We compare representations taken from different token positions. Specifically, we consider (i) the final token of the input prompt and (ii) the final token of the model's generated output. Because the final output token attends to the full prompt and any intermediate reasoning (Vaswani et al., 2023), it can provide a more comprehensive summary signal. For this ablation, we use the last layer representation and apply no pooling in both cases. Table 9 shows that using the last output token yields substantially better discrimination and calibration than using the last input token. We therefore use the final output-token representation by default.

**Pooling strategy.** We evaluate whether aggregating information across tokens improves trust prediction. We compare 1) using only the final output token hidden state (no pooling) to 2) mean pooling over hidden states across all output tokens. In both settings, we use the last layer representations and restrict pooling to output tokens. Table 10 shows that mean pooling substantially degrades performance relative to using the final token representation. We therefore use no pooling by default.

### A.2. Data Ablations

We ablate the amount of labeled data used to train the neural trust function. We train a 4-layer MLP trust function with hidden width 512 on varying numbers of chess training examples and evaluate on a disjoint set of 1,000 chess examples, using `Qwen3-0.6B` as the weak teacher. Across settings, we keep the NTF's architecture, optimization setup, and input representation fixed (last layer, last output token, no pooling), and vary only the training set size.

Table 11 shows that performance improves as the training set grows, with AUC increasing and Brier decreasing across scales. Even 1,000 examples yield a usable trust function, while 50,000 examples provide the strongest overall performance. Calibration is best at 10,000 examples, and slightly worsens at 50,000, suggesting that additional data primarily improves discrimination. Overall, neural trust functions achieve strong performance with relatively modest training data, indicating that these lightweight NTFs are not especially data-hungry.

### A.3. Student Training Budget Ablation

We also ablate the downstream student training budget $n$, i.e., the number of selected target examples used to train the student. This differs from the preceding NTF data ablation, which varies the labeled source data used to train the trust function. For each $n$, we compare NTF-retained weak supervision against a budget-matched ground-truth baseline

| $n$ | AUC ($\uparrow$) | ECE ($\downarrow$) | Brier ($\downarrow$) |
|---|---|---|---|
| 100 | 0.7588 | 0.0489 | 0.1903 |
| 1,000 | 0.8423 | 0.0351 | 0.1549 |
| 10,000 | 0.8903 | **0.0104** | 0.1380 |
| 50,000 | **0.9562** | 0.0197 | **0.0801** |

*Table 11.* **Data ablation for neural trust functions on chess using `Qwen3-0.6B`.** We vary the number of training examples used to fit the trust function and report evaluation metrics on a fixed held-out set of 1,000 examples.

| $n$ | Qwen3-1.7B NTF | Qwen3-1.7B GT | Qwen3-4B NTF | Qwen3-4B GT |
|---|---|---|---|---|
| 500 | 0.0000 | 0.0000 | 0.0137 | 0.0069 |
| 1,000 | 0.2402 | 0.2127 | 1.7361 | 1.5988 |
| 5,000 | 3.4379 | 1.6606 | 5.7504 | 4.0211 |
| 10,000 | 17.1070 | 18.6166 | 30.1585 | 28.3538 |
| 25,000 | 27.4137 | 23.7906 | 33.5003 | 38.6743 |
| 50,000 | 25.4000 | 23.0000 | 35.4000 | 36.3000 |

*Table 12.* **Student training budget ablation in strategy games.** NTF remains competitive with budget-matched ground-truth supervision across a range of training set sizes and outperforms GT in 8 of 12 model-budget combinations.

in strategy games.

Table 12 shows that NTF-retained weak supervision remains competitive with budget-matched ground-truth supervision across student training budgets. At the smallest budget, both approaches yield near-zero performance, suggesting that 500 examples are insufficient for effective student training in this setting. As the budget increases, NTF provides clear gains in several low-and-mid-budget regimes, outperforming the GT baseline for both student sizes at $n = 1,000$ and $n = 5,000$. At larger budgets, the comparison becomes more mixed, with each supervision source achieving the stronger result in different model-budget settings. Overall, these results indicate that NTF filtering can recover much of the benefit of ground-truth supervision, and often exceed a budget-matched GT baseline, particularly when training data is limited.

### A.4. Dimensionality Ablations

Our neural trust functions take as input a teacher hidden representation $h \in \mathbb{R}^D$ (§2). A natural question is whether the correctness signal that supports trust scoring is genuinely high-dimensional, or whether it is largely contained in a lower-dimensional subspace. If the latter holds, then (i) trust estimation may be robust to aggressive compression, and (ii) the features used by the trust function may reflect a small number of shared directions in representation space.

We test this by applying principal component analysis (PCA) to teacher embeddings and training NTFs on progressively lower-dimensional representations. To isolate the effect of dimensionality alone, we fix all design choices to the

| $d$ | AUC ($\uparrow$) | ECE ($\downarrow$) | Brier ($\downarrow$) | EVR ($\uparrow$) |
|---|---|---|---|---|
| 16 | 0.87 | 0.03 | 0.14 | 0.653 |
| 32 | 0.90 | 0.03 | 0.12 | 0.785 |
| 64 | 0.91 | 0.02 | 0.11 | 0.884 |
| 128 | 0.92 | 0.02 | 0.11 | 0.942 |
| 256 | 0.93 | 0.03 | 0.10 | 0.975 |
| 512 | 0.93 | 0.03 | 0.10 | 0.993 |
| 1024 | 0.91 | 0.02 | 0.11 | — |

*Table 13.* **Dimensionality ablation via PCA.** We train NTFs on PCA-compressed teacher representations of varying dimension $d$ and report discrimination, calibration, and explained variance on the held-out evaluation split.

best setting identified in prior ablations: $n$=50,000 training examples, last-layer representations, final output-token position, and no pooling. Concretely, we fit a PCA map on the *training* embeddings only,

$$\tilde{h}_i^{(k)} = \text{PCA}_k(h_i) \in \mathbb{R}^k,$$

and apply the same fitted transform to test embeddings to avoid leakage. We then train an NTF on $\tilde{h}^{(k)}$ while holding all other choices and hyperparameters fixed. We sweep $k \in \{8, 16, 32, 64, 128, 256, 512\}$, capped at $D = 1024$, and report the standard metrics (AUC, ECE, Brier) on the held-out evaluation split. We also report the cumulative explained variance ratio (EVR). In this ablation, we use the original evaluation set used in §2, instead of the sub-sampled evaluation set used in prior ablations.

Table 13 shows that the teacher representations are highly compressible in variance terms (EVR reaches 0.884 at $d$=64 and 0.942 at $d$=128), and, importantly, that the trust signal remains largely intact under substantial compression. AUC improves monotonically from 0.87 at $d$=16 to 0.93 at $d{\geq}256$, while Brier steadily decreases from 0.14 to 0.10; ECE remains low throughout (0.02–0.03). Notably, performance at $d$=256–512 matches or slightly exceeds the full-dimensional baseline ($d$=1024), indicating that correctness-relevant information is concentrated in a moderate-dimensional subspace and that the discarded low-variance directions are not necessary for (and may even mildly hinder) trust estimation. Overall, these results suggest that neural trust functions primarily rely on a lower-dimensional subspace of teacher activations, and that aggressive dimensionality reduction can preserve (or slightly improve) both discrimination and calibration. Yet, we still use the full-dimension baseline throughout the paper.

### A.5. Other Ablations

In addition to the ablations above, we explore several natural variants of neural trust functions in the MCQA training interface. These experiments are primarily empirical. Because none of these ablations produce consistent improvements, we report them qualitatively.

| Domain | Regime | Teacher | NTF Train Set | NTF Eval Set | AUC (↑) | ECE (↓) | Brier (↓) | Purity (↑) |
|---|---|---|---|---|---|---|---|---|
| World Knowledge | OOD$_{dist}$ | OLMo2-1B | MMLU$_{train}$ | ARC-CHALLENGE$_{train}$ | 0.996 | 0.010 | 0.011 | 0.981 |
| | | | | ARC-EASY$_{train}$ | 0.998 | 0.005 | 0.008 | 0.990 |
| | | | | OPENBOOKQA$_{train}$ | 1.000 | 0.007 | 0.001 | 1.000 |
| | | | | SCIQ$_{train}$ | 0.781 | 0.064 | 0.192 | 0.920 |
| | | | | SOCIALIQA$_{train}$ | 0.689 | 0.107 | 0.229 | 0.755 |
| | | | | **Avg.** | **0.893** | **0.039** | **0.088** | **0.929** |
| | | Qwen3-0.6B | MMLU$_{train}$ | ARC-CHALLENGE$_{train}$ | 0.998 | 0.019 | 0.006 | 0.995 |
| | | | | ARC-EASY$_{train}$ | 0.997 | 0.010 | 0.007 | 0.996 |
| | | | | OPENBOOKQA$_{train}$ | 1.000 | 0.005 | 0.000 | 1.000 |
| | | | | SCIQ$_{train}$ | 0.851 | 0.072 | 0.145 | 0.980 |
| | | | | SOCIALIQA$_{train}$ | 0.761 | 0.022 | 0.199 | 0.905 |
| | | | | **Avg.** | **0.921** | **0.026** | **0.071** | **0.975** |
| Quantitative Reasoning | OOD$_{dist}$ | Qwen3-1.7B | MATH$_{train}$ | OMNI-MATH | 0.830 | 0.114 | 0.130 | 0.685 |
| | | Qwen3-4B | MATH$_{train}$ | OMNI-MATH | 0.814 | 0.076 | 0.148 | 0.676 |
| | | Qwen3-8B | MATH$_{train}$ | OMNI-MATH | 0.809 | 0.098 | 0.158 | 0.726 |
| | | Gemma3-1B | MATH$_{train}$ | OMNI-MATH | 0.841 | 0.136 | 0.183 | 0.954 |
| Strategy Games | ID | OLMo2-1B | LICHESS$_{train}$ | LICHESS$_{validation}$ | **0.930** | **0.017** | **0.086** | **0.930** |
| | | Qwen3-0.6B | LICHESS$_{train}$ | LICHESS$_{validation}$ | **0.914** | **0.022** | **0.113** | **0.953** |

*Table 14.* **Full evaluation results of neural trust functions (NTFs).** We report discrimination (AUC), calibration (ECE, Brier), and *purity* of the retained subset.

**NTF pretraining from next-token prediction.** We investigate whether a trust function could be pretrained at scale using binary correctness labels derived from the teacher's next-token prediction behavior, with the goal of learning a general-purpose correctness signal that transfers across domains. We train on roughly $10^6$ such instances, then evaluate transfer under OOD$_{domain}$ (NTP → MCQA). However, we find that this practice does not yield a reliable ID and OOD$_{domain}$ performance.

**Using embeddings from all answer choices.** Our default MCQA trust function scores only the teacher representation corresponding to the predicted (selected) answer choice. We also try incorporating representations for *all* answer choices, e.g., running the NTF on each option embedding and treating the highest-scoring option as the trusted choice (with others implicitly untrusted). However, we find that this formulation does not outperform the simpler single-choice variant, while increasing compute by approximately $4\times$ due to scoring every option.

**Generative MCQA interface with explicit reasoning.** We convert MCQA into a generative setting where the model first produces a short reasoning trace and then outputs an answer choice. This modification does not meaningfully improve downstream performance relative to the standard discriminative MCQA setup, consistent with the idea that (i) the additional generation introduces extra variance, and (ii) our trust function already captures the most salient correctness cues from the teacher's internal state without requiring explicit reasoning text.

### A.6. Full Evaluation Results of NTFs

Table 1 reports selected evaluation results of NTFs. We provide the evaluation results of each NTF used for each teacher in each setting and benchmark in Table 14.

### A.7. Detailed NTF Architectures

We provide the exact NTF architectures and optimization hyperparameters used in each domain and teacher setting in Table 15, which summarizes the depth, width, dropout, learning rate, weight decay, and class reweighting coefficient used for each benchmark. We select these hyperparameters via a grid search and then fix them for all downstream evaluations in a domain.

## B. Generalization Under Different Regimes

We evaluate neural trust functions under three generalization regimes: (i) *in-distribution* (ID), (ii) *in-domain distribution shift* (OOD$_{dist}$), and (iii) *out-of-domain* (OOD$_{domain}$). We aim to understand how trust signals transfer as the relationship between training and evaluation data changes.

Table 16 reports results averaged over five MCQA benchmarks (ARC-E, ARC-C, OBQA, SCIQ, SIQA). In the **ID** setting, where $\tau$ is trained and evaluated within the same benchmark, neural trust function achieves solid discrimination (AUC = 0.8171) with very strong calibration (ECE = 0.0152, Brier = 0.1672). Under **OOD$_{dist}$**, where $\tau$ is trained on MMLU and evaluated zero-shot on the MCQA benchmarks, performance remains strong and in fact improves on average (AUC = 0.9214, Brier = 0.0714). We at-

| Domain | Teacher | Depth and Width | Dropout | LR | WD | Class RW |
|---|---|---|---|---|---|---|
| World Knowledge | OLMo2-1B | 4×512 | 0.2 | $5\times10^{-5}$ | 0.0001 | 1.0 |
| | Qwen3-0.6B | 4×512 | 0.2 | $5\times10^{-5}$ | 0.0001 | 0.5 |
| Quantitative Reasoning | Qwen3-1.7B | 8×1024 | 0.2 | $1\times10^{-5}$ | 0.001 | 1.0 |
| | Qwen3-4B | 8×1024 | 0.2 | $1\times10^{-5}$ | 0.001 | 1.0 |
| | Qwen3-8B | 8×1024 | 0.2 | $1\times10^{-5}$ | 0.001 | 1.0 |
| | Gemma3-1B | 8×1024 | 0.2 | $1\times10^{-5}$ | 0.001 | 1.0 |
| Strategy Games | OLMo2-1B | 4×512 | 0.2 | $1\times10^{-5}$ | 0.0001 | 0.5 |
| | Qwen3-0.6B | 4×512 | 0.2 | $1\times10^{-5}$ | 0.0001 | 0.5 |

*Table 15.* **NTF architecture and optimization settings by domain and teacher.**

| Regime | AUC ($\uparrow$) | ECE ($\downarrow$) | Brier ($\downarrow$) |
|---|---|---|---|
| ID | 0.8171 | **0.0152** | 0.1672 |
| $OOD_{dist}$ | **0.9214** | 0.0256 | **0.0714** |
| $OOD_{domain}$ | 0.5560 | 0.0337 | 0.2351 |

*Table 16.* **Generalization of neural trust functions across regimes on MCQA benchmarks.** $OOD_{dist}$ preserves strong discrimination and calibration, while $OOD_{domain}$ transfer collapses.

| Method | Teacher: Gemma3-1B LLaMA3.1-8B |
|---|---|
| Teacher Performance | **43.52** |
| No-GRPO | 45.76 |
| Naive | 42.96 (-47.0) |
| I-Confidence | 41.32 (-74.5) |
| V-Confidence | 44.24 (-25.5) |
| **NTF (Ours)** | **51.08** (92.2) |
| Ground Truth | 51.72 |

*Table 17.* **Transfer results on MATH using Gemma3-1B supervision to train LLaMA3.1-8B.** Parentheses report recovery. NTF cell colors follow the significance legend from Table 2 (see App. G).

tribute this gain primarily to scale: MMLU provides one to two order of magnitude more labeled examples than the individual MCQA training sets (100K vs. typically 1–10K), enabling $\tau$ to learn a more stable and transferable correctness signal. This interpretation is consistent with the monotonic improvements we observe as NTF training data increases in App. A.2.

In contrast, **$OOD_{domain}$** transfer exhibits a sharp degradation. When trained in the strategy games domain and evaluated in the world knowledge domain, the trust function's discrimination drops substantially (AUC = 0.5560) and its overall error increases (Brier = 0.2351). This suggests that the learned trust signal depends on domain-specific representational structure and does not reliably transfer across fundamentally different task interfaces.

Overall, these results reinforce our core distinction: neural trust functions transfer well under in-domain distribution shifts ($OOD_{dist}$) but degrade under out-of-domain transfer ($OOD_{domain}$), motivating our focus on the former regime throughout the paper.

## C. Addressing the Spurious Reward Concern

A common concern in RL-based post-training is the *spurious reward* phenomenon, which has been observed specifically for QWEN-family models (Shao et al., 2025). In our main experiments, we focus on demonstrating performance gains on difficult mathematical reasoning tasks; empirically,

only Qwen3 base models are sufficiently capable to fit these datasets, despite starting from very low initial accuracy. To verify that the improvements reported above are not merely artifacts of the spurious reward phenomenon, and to provide a stronger comparison against baselines, we train Llama3.1-8B-Instruct, a model family that (Shao et al., 2025) reports as substantially less susceptible to spurious reward than Qwen model family. In this setting, we apply our L2T framework with Gemma3-1B as the weak teacher and Llama3.1-8B-Instruct as the strong student. We train the neural trust functions on GSM8K (Cobbe et al., 2021), use the MATH training set (Hendrycks et al., 2021b) (approximately 7,500 problems) as the unlabeled pool for trust filtering, and evaluate on MATH-500 (Lightman et al., 2023). We select the top 500 MATH training examples ranked by the NTF for student training.

Table 17 shows that NTF remains effective even outside the Qwen3 model family in the quantitative reasoning domain, where NTF improves performance to 51.08, substantially outperforming all baselines and nearly matching the ground-truth performance with 92.2% recovery. Since LLaMA3.1-8B is reported to be substantially less susceptible to spurious rewards than Qwen models, these results

indicate that the utility of NTFs in the quantitative reasoning domain cannot be explained solely by the spurious reward phenomenon.

# D. Training Details

## D.1. World Knowledge

**Overview.** For world knowledge, we fine-tune student models using LoRA-based supervised fine-tuning (SFT). To ensure that comparisons reflect *selection quality* rather than differences in data quantity, we match the training set size to an oracle budget derived from the teacher's accuracy on the target training split.

**Budget definition.** Given a target dataset with training inputs $\{x_i\}_{i=1}^N$, we first run the weak teacher $\pi_{\mathcal{W}}$ to obtain predictions $\hat{y}_i$. For *evaluation only*, we define a supervision budget that matches the expected number of correct weak labels available in the pool by using the ground-truth labels on the target training split *solely to set* $n$:

$$n = \sum_{i=1}^{N} \mathbf{1}\{\hat{y}_i = y_i\}.$$

This oracle budgeting ensures that all selection rules are compared under the same *effective* supervision amount (i.e., the same upper bound on usable weak labels), so performance differences reflect *which* examples are selected rather than how many correct labels happen to be included. To control training cost and keep budgets comparable across datasets, we cap this value at 2000 and use $n = \min(n, 2000)$. Crucially, ground truth is *never* used for selection, as we use it only to define a common budget for controlled comparisons. In practical settings, $n$ can be set using a small labeled calibration subset or other label-free budgeting heuristics (App. I).

**Trust filtering.** We then apply the neural trust function to each teacher-labeled example to obtain trust scores $t_i = \tau(g_{\pi_{\mathcal{W}}}(x_i, \hat{y}_i))$. We form the weakly supervised training set by retaining the top-$n$ examples ranked by $t_i$:

$$\tilde{\mathcal{D}} = \{(x_i, \hat{y}_i) \text{ among the top } n \text{ by } t_i\}.$$

Intuitively, this procedure approximates an oracle that retains exactly the teacher-correct subset, while remaining supervision-free at selection time.

**Student training.** Finally, we fine-tune the student $\pi_{\mathcal{S}}$ on $\tilde{\mathcal{D}}$ using LoRA-SFT with the same hyperparameters across selection methods. Students are evaluated on the target test split using standard multiple-choice accuracy. All training runs are implemented in VeRL (Sheng et al., 2025) and use 2×A100 80GB GPUs with gradient checkpointing and accumulation enabled, fp32 precision, and FlashAttention-2 (Dao, 2023). Since the teacher outputs only the final answer

| Hyperparameter | Value |
|---|---|
| Max seq. length | 1024 |
| Global batch size | 128 |
| Micro batch size (per GPU) | 4 |
| Epochs | 3 if $n < 2000$, else 1 |
| Precision | fp32 |
| Attention | FlashAttention-2 |
| Optimizer | AdamW |
| Learning rate | $1 \times 10^{-4}$ |
| Adam $\beta$ | (0.9, 0.95) |
| Weight decay | 0.01 |
| LR scheduler | Cosine (Loshchilov & Hutter, 2017) |
| Warmup ratio | 0.1 |
| Gradient clip norm | 1.0 |
| LoRA target modules | all-linear |
| LoRA rank ($r$) | 16 |
| LoRA $\alpha$ | 32 |
| LoRA dropout | 0.0 |

*Table 18.* **SFT hyperparameters used for student training in the world knowledge domain.**

choice (no intermediate reasoning), SFT updates are applied only to the answer-choice tokens. Optimization and LoRA hyperparameters are provided in Table 18.

## D.2. Quantitative Reasoning

We study mathematical reasoning under several supervision regimes. For **ground-truth supervision**, we use the dataset-provided correctness labels. For **weak supervision**, we derive labels from model rollouts using four alternatives: *NTF*, *V-Confidence*, *I-Confidence*, and a *Naive* baseline. All models are trained with **GRPO** using the corresponding labels.

**NTF training and calibration.** For the Qwen3 family, we train the NTF on the MATH training split and calibrate it on the MATH test split. At filtering time, we apply the NTF to the *last-layer* hidden states.

**Data filtering on OMNI-MATH.** To avoid confounding selection quality with data quantity, we hold the number of retained weakly labeled examples fixed across all selection methods for a given model. We first estimate an oracle upper bound on the number of usable positive examples by counting how many OMNI-MATH questions the base model answers correctly in a single-pass evaluation. Since a learned selector cannot identify this entire set perfectly, we convert this oracle count into a more conservative retention budget using the estimated purity of the NTF. Specifically, if $n_{\text{correct}}$ denotes the number of base-model rollouts that are correct and $\hat{p}_{\text{NTF}}$ denotes the NTF's purity on its training

set , we set

$$k = \lfloor \widehat{p}_{\text{NTF}} \cdot n_{\text{correct}} \rfloor .$$

This gives all weak-supervision methods the same effective retention budget, while accounting for the fact that NTF-based filtering is imperfect. Thus, performance differences primarily reflect the quality of the selected examples rather than the number of examples used for GRPO training.

We use OMNI-MATH as the pool to be filtered. For each example, we generate an answer with temperature 0.0 and a maximum generation length of 4096. We then score examples with each selection method and retain its top-$k$ subset using the budget defined above. When our default math parser cannot verify an answer, we fall back to the Omni-Judge released by the dataset authors to adjudicate correctness.

**Distribution-Aware Random Sampling.** To avoid the instability of pure random sampling, we apply a distribution-aware sampling strategy when constructing the random baseline. Each example in the dataset is annotated with a difficulty score and an associated correctness flag (is_correct) indicating whether the model answers the question correctly. We first partition the data into difficulty bins, compute the model's accuracy within each bin, and use this accuracy signal to derive sampling weights. The total sampling budget is then allocated across bins proportionally to these weights, ensuring that the resulting subset reflects a controlled difficulty composition informed by model performance. Finally, we uniformly sample within each bin according to its assigned quota. This procedure produces a more representative and stable baseline compared to unconstrained random selection.

**Implementation details.** We remove the chat template during training and keep the GRPO hyperparameters fixed across all runs (Table 19). All training runs are implemented in VeRL (Sheng et al., 2025) and use 4×A100 80GB GPUs with gradient checkpointing and accumulation enabled, fp32 precision, and FlashAttention-2.

For evaluation, we report the mean accuracy over 5 runs. We use the following decoding parameters: max_tokens=4096, temperature=0.6, top_p=0.8, top_k=20, min_p=0.0, and presence_penalty=1.5.

### D.3. Strategy Games

For strategy games, we fine-tune student models using LoRA-based supervised fine-tuning (SFT). Unless otherwise stated, we use the same training pipeline and hyperparameters as in the world knowledge setting (App. D.1). In contrast to world knowledge, where the training budget is derived from the teacher's accuracy on the target training split, we use a fixed supervision budget of $n = 50{,}000$

| Hyperparameter | Value |
|---|---|
| Batch size | 128 |
| Max prompt length | 1024 |
| Max response length | 4096 |
| Filter overlong prompts | True |
| Learning rate | $1 \times 10^{-6}$ |
| Remove padding | True |
| KL loss | True |
| KL loss coefficient | 0.001 |
| Entropy coefficient | 0 |
| KL in reward | False |
| Critic warmup | 0 |
| PPO mini batch size | 128 |
| PPO micro batch size (per GPU) | 8 |
| Log prob micro batch size (per GPU) | 8 (rollout/ref) |
| Epochs | 20 |

*Table 19.* **GRPO hyperparameters used for student training in the quantitative reasoning domain.**

puzzles for all strategy games experiments. This choice standardizes training cost across teachers and students and isolates the effect of selection quality. Accordingly, we train the students for a single epoch.

**Format adaptation.** In this domain, base (pretrained) checkpoints achieve near-zero accuracy in our *generative* evaluation setting (§E.3) and often fail to produce a valid move string, yielding degenerate weak supervision. Therefore, we generate weak labels $\hat{y}$ and extract teacher hidden states using a *lightly SFT'ed* version of each weak teacher. We briefly LoRA-SFT these teachers on a *disjoint* labeled puzzle subset (10K examples) (no overlap with any student training or evaluation puzzles) to ensure non-degenerate move generation and non-trivial accuracy. This teacher SFT uses the same prompt format, pipeline, and hyperparameters as student SFT (App. D.1), but is run for a short duration and is used *only* for label and embedding generation. Teacher Performance refers to the accuracy of this lightly SFT'ed teacher, while No-SFT baseline is 0.0% for all models and is omitted. Thus, Recovery is computed with Base = 0.

## E. Evaluation Details

### E.1. World Knowledge

We evaluate multiple-choice question answering by scoring each candidate option under the model. Concretely, for each example we compute the conditional log-probability of each

answer option given the question prompt, and predict the option with maximum log-likelihood. All evaluations are performed in a zero-shot setting (no in-context demonstrations), with the only exception being **ICL + I-Confidence**, which prepends a fixed set of in-context examples to the prompt when computing option scores.

### E.2. Mathematical Problem Solving

We evaluate mathematical reasoning in a zero-shot setting using the following generation parameters: temperature $= 0.6$, max tokens $= 4096$, presence penalty $= 1.5$, top-$p$ $= 0.8$, and top-$k = 20$. For answer matching, we extract the final boxed expression from each model output and compare it against the ground-truth answer using `latexsympy`. For OMNI-MATH, we additionally employ an LLM-as-Judge protocol via Omni-Judge, released together with the dataset (Gao et al., 2024). For MATH (Hendrycks et al., 2021b), we only use `latexsympy` for parsing and matching. For hidden-state extraction, we do not apply the chat template during evaluation; however, after training, we apply the chat template for testing. Finally, for instruction-tuned models `Gemma3-1B` and `Llama3.1-8B`, we apply the chat template in both hidden-state extraction and evaluation.

### E.3. Strategy Games

We evaluate chess puzzles in a *generative* setting, where the model is prompted to produce the next move from a textual description of the current board state. Each position is serialized as structured metadata followed by a piece list for both sides. Concretely, the prompt includes (i) the side to move, (ii) castling rights, (iii) en-passant availability, (iv) halfmove and fullmove counters, and (v) a list of pieces with their squares, e.g.:

> Side to move: Black
> Castling rights: -
> En passant: -
> Halfmove: 3 Fullmove: 21
> White: a1 Rook; c1 Bishop; g1 King; a2 Pawn;
> b2 Pawn; c2 Pawn; g2 Pawn; h2 Pawn; d3 Queen
> Black: d4 Pawn; c5 Pawn; e5 Queen; a7 Pawn;
> g7 Pawn; h7 Pawn; f8 Rook; g8 King

The model outputs a single candidate move, which we score as correct if it matches the puzzle's labeled best move. Because generation quality can be sensitive to decoding, we sweep a small grid of sampling hyperparameters and report the best accuracy for each model. Concretely, we evaluate (i) greedy decoding (`temperature = 0`), (ii) nucleus/top-$k$ sampling with `temperature = 0.7`, `top_k = 20`, `top_p = 0.8`, and (iii) higher-temperature sampling with `temperature = 1.0`, `top_k = -1` (no limit), `top_p = 1.0`.

| Teacher → | `Qwen3-1.7B` | | `Qwen3-4B` | `Qwen3-8B` |
|---|---|---|---|---|
| Method ↓ – Student → | `Qwen3-4B` | `Qwen3-8B` | `Qwen3-8B` | `Qwen3-8B` |
| Reward Model | 23.3 | — | 27.5 | — |
| **NTF (Ours)** | 22.0 | 26.6 | 27.9 | 27.4 |
| Ground Truth | 22.9 | 27.4 | 28.7 | 28.4 |

*Table 20.* **Math reward-model baseline.** Reward Model uses Qwen2.5-Math-RM-72B to rank weak teacher rollouts under the same selection budget.

## F. Baseline Details

We compare neural trust functions against a common set of weak-to-strong supervision baselines that span (i) *uninformed selection*, (ii) *output-level uncertainty signals*, (iii) *self-reported confidence*, (iv) *multi-teacher agreement*, (v) *external verifier models*, and (vi) *ground-truth supervision*. Unless otherwise stated, all methods use the same supervision budget (number of selected training examples) so differences reflect *selection quality* rather than data quantity. **No-SFT & No-GRPO.** The student $\pi_S$ is evaluated zero-shot without any additional training on $\mathcal{D}_u$.

**Naive.** We train the student on weak labels from $n$ examples selected without any filtering. For world knowledge and strategy games, we sample uniformly at random from the weakly labeled pool. For quantitative reasoning, we use difficulty-stratified random sampling (sampling within each difficulty bucket). This baseline isolates the effect of using weak supervision *per se*, producing training data whose label quality is, in expectation, comparable to the weak teacher's accuracy.

**Internal Confidence (I-Confidence).** We rank examples by an output-level confidence score derived from the weak teacher $\pi_W$ and retain the top-$n$. Concretely, we use a length-normalized log-probability of the teacher's predicted label (instantiated per domain; see the corresponding results subsection). **In-Context Learning + Internal Confidence (ICL + I-Confidence).** Identical to **I-Confidence**, except we prepend a fixed set of five in-context demonstrations to the prompt before computing confidence and selecting the top-$n$ examples.

**Verbalized Confidence (V-Confidence).** We prompt the teacher to explicitly report a confidence score (or probability) for its predicted label and select the top-$n$ by this self-reported confidence. This baseline captures *verbalized* uncertainty as an alternative to log-probability-based confidence.

**Ensemble.** We use two independent weak teachers and select examples where their predictions agree. This baseline measures the benefit of additional teacher queries and inter-model consistency. Due to limited available weak teachers, we do not consider this baseline in the quantitative reasoning

| Model | Domain | $A$ | $p$-value |
|---|---|---|---|
| OLMo2-1B | World Knowledge | NTF | 0.0253 |
| OLMo2-7B | World Knowledge | Ground Truth | 0.5000 |
| OLMo2-13B | World Knowledge | Ground Truth | 0.0260 |
| Qwen3-0.6B | World Knowledge | Ground Truth | 0.3474 |
| Qwen3-1.7B | World Knowledge | NTF | 0.5000 |
| Qwen3-4B | World Knowledge | Ground Truth | 0.0324 |
| Qwen3-8B | World Knowledge | Ground Truth | 0.2398 |
| Qwen3-14B | World Knowledge | NTF | 0.1298 |
| OLMo2-1B | Strategy Games | Ground Truth | 0.1972 |
| OLMo2-7B | Strategy Games | Ground Truth | $4.50 \times 10^{-188}$ |
| OLMo2-13B | Strategy Games | Ground Truth | $4.96 \times 10^{-194}$ |
| Qwen3-0.6B | Strategy Games | NTF | 0.0290 |
| Qwen3-1.7B | Strategy Games | NTF | $1.04 \times 10^{-8}$ |
| Qwen3-4B | Strategy Games | Ground Truth | 0.0073 |
| Qwen3-8B | Strategy Games | NTF | 0.0015 |
| Qwen3-14B | Strategy Games | NTF | $2.49 \times 10^{-19}$ |
| Qwen3-4B (Teacher: Qwen3-1.7B) | Quantitative Reasoning | Ground Truth | 0.0120 |
| Qwen3-8B (Teacher: Qwen3-1.7B) | Quantitative Reasoning | Ground Truth | 0.1053 |
| Qwen3-8B (Teacher: Qwen3-4B) | Quantitative Reasoning | Ground Truth | 0.1024 |
| Qwen3-8B (Teacher: Qwen3-8B) | Quantitative Reasoning | Ground Truth | 0.0410 |
| Llama3.1-8B (Teacher: Gemma3-1B) | Quantitative Reasoning | Ground Truth | 0.2631 |

*Table 21.* **One-sided exact paired significance tests.**

domain.

**Reward Models.** We compare NTF against public reward-model and verifier baselines. For world knowledge, we use ArmoRM-Llama3-8B-v0.1 (Wang et al., 2024), Skywork-Reward-V2-Qwen3-8B (Liu et al., 2026), and CompassVerifier-3B (Liu et al., 2025), selecting the strongest RM/verifier for each benchmark under the same top-$n$ selection budget. For quantitative reasoning, we additionally evaluate Qwen2.5-Math-RM-72B (Yang et al., 2024a) as a math-specialized reward-model baseline. Since this RM is substantially larger and specialized for mathematical reasoning, we report it separately in Table 20 rather than in the main text.

**Ground Truth.** We train the student on $n$ ground-truth labeled examples from the target domain. This provides a budget-matched oracle reference for the best achievable performance under standard supervised fine-tuning.

**NTF (Ours).** We compute a trust score $t = \tau(g_{\pi_{\mathcal{W}}}(x, \hat{y}))$ from the teacher's internal representation and select the top-$n$ examples by $t$. NTF is single-pass (one teacher forward per example) and does not require additional sampling, self-evaluation prompts, or multiple teachers.

Note that all baselines operate under the same training pipeline and hyperparameters for a given domain, differing only in how training examples are selected (or weighted) from $\tilde{\mathcal{D}}_u$.

## G. Statistical Significance Tests

We assess whether observed differences between **NTF** and **Ground Truth** are statistically significant using an *exact paired test* on per-instance correctness. For each setting, let $A$ denote the variant with higher empirical accuracy among {**NTF**, **Ground Truth**}, and let $B$ denote the other one. We form the paired $2 \times 2$ table and summarize it by the discordant counts: $n_{AB}$ (instances where $A$ is correct and $B$ is incorrect) and $n_{BA}$ (instances where $A$ is incorrect and $B$ is correct). Under the null hypothesis that the two variants have equal marginal accuracy, discordant outcomes are symmetric, so

$$n_{AB} \sim \text{Binomial}(n_{AB} + n_{BA}, 1/2).$$

We report the *one-sided* exact $p$-value for the alternative $A > B$ (equivalently, $n_{AB} > n_{BA}$). Throughout the paper, we use $\alpha = 0.05$ to determine statistical significance. Table 21 lists the resulting $p$-values for each setting, along with which variant is $A$.

## H. Difficulty Distribution in Quantitative Reasoning

We summarize the distribution of NTF filtered difficulties over the OMNI-MATH dataset (difficulty level 1 to 5) in Table 22. The Acc column refers to the accuracy parsed by `latexparser`. The Omni-Judge refers to the remaining

questions judged as incorrect by `latexparser`, and how many are labeled as correct by the Omni-Judge. We observe the general pattern of selecting less difficult questions across all models.

## I. Risk-Controlled Data Selection

The main experiments use a fixed, budget-matched top-$n$ selection protocol so that all methods train on the same number of target examples. In practical deployments, however, the user may not know the correct retention budget. We therefore describe a risk-controlled calibration procedure for choosing either a score threshold or a top-$k$ retention rule from a small labeled target calibration subset.

**Setup.** Let $s(x, \hat{y}) \in [0, 1]$ denote the trust score assigned to a weakly labeled example $(x, \hat{y})$, where larger scores indicate greater confidence that $\hat{y}$ is correct. For a candidate threshold $\theta$, define the selected set

$$S(\theta) = \{(x, \hat{y}) : s(x, \hat{y}) \geq \theta\}.$$

On a labeled calibration subset, let $z_i = 1$ indicate that the weak label is correct and $z_i = 0$ otherwise. The empirical noise rate among selected calibration examples is

$$\widehat{r}(\theta) = \frac{1}{n(\theta)} \sum_{i:s_i \geq \theta} (1 - z_i), \qquad n(\theta) = |\{i : s_i \geq \theta\}|.$$

**Upper confidence bound and selection rule.** To avoid selecting thresholds using empirical noise alone, we compute an upper confidence bound on the true noise rate. Using Hoeffding's inequality (Hoeffding, 1963), for each candidate threshold $\theta$ we define

$$U(\theta; \delta_\theta) = \widehat{r}(\theta) + \sqrt{\frac{\log(1/\delta_\theta)}{2n(\theta)}}.$$

Given a user-specified target noise rate $\alpha$, we select the most inclusive threshold satisfying

$$\theta^\star = \min\{\theta : U(\theta; \delta_\theta) \leq \alpha\}.$$

**Multiple-testing correction.** For a fixed threshold $\theta$, Hoeffding's inequality gives

$$\Pr[r(\theta) \leq U(\theta; \delta_\theta)] \geq 1 - \delta_\theta,$$

where $r(\theta)$ is the true noise rate among examples selected by $\theta$. Since we choose $\theta^\star$ after evaluating $m$ candidate thresholds, we use a Bonferroni correction and set $\delta_\theta = \delta/m$. Then, by a union bound, with probability at least $1 - \delta$, all candidate thresholds satisfy

$$r(\theta) \leq U(\theta; \delta/m).$$

Therefore, any selected threshold with $U(\theta; \delta/m) \leq \alpha$ has true noise rate at most $\alpha$.

In our chess setting with $n_{\text{cal}} = 3000$, $\alpha = 0.1$, $\delta = 0.1$, and $m = 3000$ unique calibration scores, this correction is overly conservative: $\delta_\theta \approx 3.3 \times 10^{-5}$ inflates the Hoeffding term enough that the best achievable UCB is $\approx 0.151 > \alpha$, making the threshold-grid procedure formally infeasible.

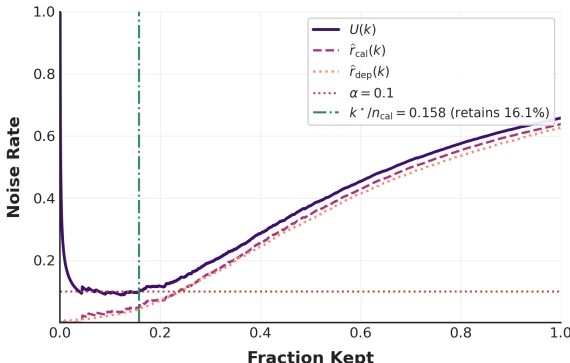

*Figure 6.* **Risk-controlled top-$k$ calibration.** Sorting calibration examples in descending order of trust score, $\widehat{r}_{\text{cal}}(k)$ is the empirical noise rate on the top-$k$ prefix, $U(k)$ is its Hoeffding upper confidence bound, and $\widehat{r}_{\text{dep}}(k)$ is the noise rate on the held-out deployment pool at the induced threshold. The largest $k$ satisfying $U(k) \leq \alpha = 0.1$ corresponds to $k^\star/n_{\text{cal}} = 0.158$, which projects to $16.1\%$ of the deployment pool, the same operating point as the threshold-mode result in Fig. 5.

We therefore report main-text results with $\delta_\theta = \delta$ (no MT correction), corresponding to a per-threshold guarantee. This is the standard choice when the threshold is treated as fixed or selected on an independent split. Two practical mitigations recover a uniform guarantee at the cost of either coarser thresholds or additional labeled data: (i) coarsening the candidate grid to a small set (e.g. $m = 100$), which keeps $\delta_\theta$ tractable; (ii) splitting the calibration data so that the threshold is chosen on one split and the bound is evaluated on the other, removing the need for correction altogether. Across all configurations we ran, the held-out deployment-pool noise rate at $\theta^\star$ stayed below $\alpha$ (Fig. 5), suggesting that the per-threshold variant generalizes in practice even though it carries a weaker formal guarantee.

**Top-$k$ variant.** We also consider a top-$k$ version. We sort calibration examples by trust score in descending order and choose the largest $k$ such that the UCB on the empirical noise rate of the top-$k$ prefix is below $\alpha$. This calibrates how much data to retain rather than directly calibrating a score threshold; the chosen $k$ induces a threshold equal to the score of the $k$-th selected example. The same MT-correction considerations apply, with $m = n_{\text{cal}}$ prefixes acting as the hypothesis family. Result shown at Fig. 6.

| Difficulty Bin | $n_{\text{all}}$ | Qwen3-1.7B Selected | | | Qwen3-4B Selected | | | Qwen3-8B Selected | | |
|---|---|---|---|---|---|---|---|---|---|---|
| | | $n$ | Acc | Omni-Judge | $n$ | Acc | Omni-Judge | $n$ | Acc | Omni-Judge |
| $[1, 2)$ | 335 | 239 | 0.7615 | 0.1754 | 223 | 0.7803 | 0.2449 | 261 | 0.8046 | 0.2157 |
| $[2, 3)$ | 343 | 106 | 0.6132 | 0.1463 | 122 | 0.6803 | 0.2051 | 146 | 0.6781 | 0.1489 |
| $[3, 4)$ | 224 | 20 | 0.6500 | 0.0000 | 41 | 0.6341 | 0.0667 | 44 | 0.6591 | 0.1333 |
| $[4, 5)$ | 957 | 37 | 0.1622 | 0.0323 | 87 | 0.3563 | 0.1429 | 57 | 0.3860 | 0.0571 |
| $[5, 6)$ | 890 | 14 | 0.2143 | 0.0000 | 70 | 0.3571 | 0.1333 | 53 | 0.4151 | 0.0645 |
| Overall Selected | 2749 | 416 | 0.6466 | 0.1156 | 543 | 0.6243 | 0.1716 | 561 | 0.6809 | 0.1341 |

*Table 22.* **Difficulty-bin breakdown of neural trust function selected high-trust positive examples across Qwen3 teachers.** We report the full dataset difficulty distribution ($n_{\text{all}}$) as reference. Each row shows the number of selected samples ($n$), empirical correctness accuracy, and Omni-Judge correctness rate.

| Referred as... | Model | Role(s) | Access Link |
|---|---|---|---|
| OLMo2-1B | allenai/OLMo-2-0425-1B | Teacher; Student | Hugging Face |
| OLMo2-7B | allenai/OLMo-2-1124-7B | Student | Hugging Face |
| OLMo2-13B | allenai/OLMo-2-1124-13B | Student | Hugging Face |
| Qwen3-0.6B | Qwen/Qwen3-0.6B-Base | Teacher; Student | Hugging Face |
| Qwen3-1.7B | Qwen/Qwen3-1.7B-Base | Teacher; Student | Hugging Face |
| Qwen3-4B | Qwen/Qwen3-4B-Base | Teacher; Student | Hugging Face |
| Qwen3-8B | Qwen/Qwen3-8B-Base | Teacher; Student | Hugging Face |
| Qwen3-14B | Qwen/Qwen3-14B-Base | Student | Hugging Face |
| Gemma3-1B | google/gemma-3-1b-it | Teacher | Hugging Face |
| Llama3.1-8B | meta-llama/Llama-3.1-8B-Instruct | Student | Hugging Face |

*Table 23.* **Model checkpoints used throughout the paper, their roles, and the access links.**

**Empirical takeaway.** Risk-controlled calibration provides a principled alternative to manual threshold tuning when the user wants a high-purity guarantee on a new target domain. On the chess benchmark, the calibrated threshold $\theta^\star = 0.895$ retains $16.1\%$ of the deployment pool, and the realized noise rate on held-out data stays below the target $\alpha = 0.1$, indicating that the trust scores are well-aligned with label correctness in the high-score regime.

## J. All Models

Table 23 lists all model checkpoints used throughout the paper, along with their roles (teacher/student) and access links.

## K. World Knowledge Full Results

### K.1. ARC-Challenge

Table 24 reports accuracy (%) in ARC-CHALLENGE for each selection rule under the two teacher settings, with all methods trained using a matched supervision budget $n$.

### K.2. ARC-Easy

Table 25 reports accuracy (%) in ARC-EASY for each selection rule under the two teacher settings, with all methods trained using a matched supervision budget $n$.

### K.3. OpenBookQA

Table 26 reports accuracy (%) in OPENBOOKQA for each selection rule under the two teacher settings, with all methods trained using a matched supervision budget $n$.

### K.4. SciQ

Table 27 reports accuracy (%) in SCIQ for each selection rule under the two teacher settings, with all methods trained using a matched supervision budget $n$.

### K.5. SocialIQA

Table 28 reports accuracy (%) in SOCIALIQA for each selection rule under the two teacher settings, with all methods trained using a matched supervision budget $n$.

| Teacher → | OLMo2-1B | | | Qwen3-0.6B | | | | |
|---|---|---|---|---|---|---|---|---|
| Method ↓ – Student → | OLMo2-1B | OLMo2-7B | OLMo2-13B | Qwen3-0.6B | Qwen3-1.7B | Qwen3-4B | Qwen3-8B | Qwen3-14B |
| Teacher Performance | 33.4 | | | 46.2 | | | | |
| No-SFT | 33.4 | 59.0 | 70.2 | 46.2 | 67.0 | 73.0 | 78.8 | 82.6 |
| Naive | 34.4 (62.5) | 62.5 (74.5) | 72.6 (85.7) | 50.7 (70.3) | 70.1 (91.2) | 77.7 (87.0) | 84.3 (78.6) | 87.2 (83.6) |
| I-Confidence | 34.3 (56.2) | 62.3 (70.2) | 72.6 (85.7) | 51.2 (78.1) | 70.5 (102.9) | 76.7 (68.5) | 83.9 (72.9) | 87.2 (83.6) |
| ICL + I-Confidence | 34.7 (81.3) | 63.6 (97.9) | 72.9 (96.4) | 52.7 (101.6) | 70.5 (102.9) | 77.4 (81.5) | 84.8 (85.7) | 87.5 (89.1) |
| Ensemble | **35.2** (112.5) | 63.4 (93.6) | 72.6 (85.7) | 48.6 (37.5) | 68.9 (55.9) | 74.3 (24.1) | 81.0 (31.4) | 84.6 (36.4) |
| Reward Model | 34.3 (56.2) | 61.6 (55.3) | 72.4 (78.6) | 52.2 (93.8) | **71.4** (129.4) | 78.0 (92.6) | 84.6 (82.9) | **87.7** (92.7) |
| **NTF (Ours)** | 34.8 (87.5) | **63.7** (100.0) | **73.2** (107.1) | **52.8** (103.1) | 71.0 (117.6) | **78.1** (94.4) | **85.1** (90.0) | **87.7** (92.7) |
| Ground Truth | 35.0 | 63.7 | 73.0 | 52.6 | 70.4 | 78.4 | 85.8 | 88.1 |

*Table 24.* **Individual results for ARC-CHALLENGE.** Accuracy (%) is reported for each teacher–student setting. Recovery (Eq. 1) for each baseline is reported inside parentheses.

| Teacher → | OLMo2-1B | | | Qwen3-0.6B | | | | |
|---|---|---|---|---|---|---|---|---|
| Method ↓ – Student → | OLMo2-1B | OLMo2-7B | OLMo2-13B | Qwen3-0.6B | Qwen3-1.7B | Qwen3-4B | Qwen3-8B | Qwen3-14B |
| Teacher Performance | 50.1 | | | 63.3 | | | | |
| No-SFT | 50.1 | 76.9 | 84.5 | 63.3 | 83.3 | 85.2 | 90.4 | 92.6 |
| Naive | 55.9 (79.5) | 81.2 (45.7) | 88.9 (58.7) | 71.0 (53.5) | 89.0 (83.8) | 92.5 (70.2) | 95.7 (82.8) | 96.7 (82.0) |
| I-Confidence | 55.6 (75.3) | 81.6 (50.0) | 88.9 (58.7) | 71.0 (53.5) | 89.3 (88.2) | 93.1 (76.0) | 95.6 (81.2) | 96.7 (82.0) |
| ICL + I-Confidence | 56.7 (90.4) | 83.5 (70.2) | 90.1 (74.7) | 76.6 (92.4) | 89.6 (92.6) | 94.3 (87.5) | 96.3 (92.2) | 97.3 (94.0) |
| Ensemble | 56.9 (93.2) | 84.6 (81.9) | 90.9 (85.3) | 73.2 (68.8) | 88.5 (76.5) | 92.9 (74.0) | 95.7 (82.8) | 97.4 (96.0) |
| Reward Model | **57.4** (100.0) | 85.1 (87.2) | 89.6 (68.0) | 75.3 (83.3) | 89.9 (97.1) | 94.5 (89.4) | 96.0 (87.5) | 97.5 (98.0) |
| **NTF (Ours)** | 57.3 (98.6) | **85.9** (95.7) | **91.9** (98.7) | **77.1** (95.8) | **90.3** (102.9) | **95.5** (99.0) | **96.9** (101.6) | **97.6** (100.0) |
| Ground Truth | 57.4 | 86.3 | 92.0 | 77.7 | 90.1 | 95.6 | 96.8 | 97.6 |

*Table 25.* **Individual results for ARC-EASY.** Accuracy (%) is reported for each teacher–student setting. Recovery (Eq. 1) for each baseline is reported inside parentheses.

| Teacher → | OLMo2-1B | | | Qwen3-0.6B | | | | |
|---|---|---|---|---|---|---|---|---|
| Method ↓ – Student → | OLMo2-1B | OLMo2-7B | OLMo2-13B | Qwen3-0.6B | Qwen3-1.7B | Qwen3-4B | Qwen3-8B | Qwen3-14B |
| Teacher Performance | 33.4 | | | 38.6 | | | | |
| No-SFT | 33.4 | 54.8 | 65.4 | 38.6 | 55.2 | 63.0 | 72.0 | 73.2 |
| Naive | 40.2 (85.0) | 55.0 (1.6) | 54.2 (-76.7) | 46.4 (84.8) | 61.0 (100.0) | 70.8 (73.6) | 78.8 (87.2) | 79.0 (85.3) |
| I-Confidence | 41.0 (95.0) | 54.6 (-1.6) | 55.4 (-68.5) | 46.4 (84.8) | 60.4 (89.7) | 71.0 (75.5) | 78.2 (79.5) | 78.4 (76.5) |
| ICL + I-Confidence | 42.0 (107.5) | 63.0 (64.1) | 67.2 (12.3) | 48.2 (104.3) | 60.4 (89.7) | 72.4 (88.7) | 79.2 (92.3) | 79.6 (94.1) |
| Ensemble | 40.6 (90.0) | 59.4 (35.9) | 61.8 (-24.7) | 49.0 (113.0) | 62.6 (127.6) | 68.4 (50.9) | 77.0 (64.1) | 77.8 (67.6) |
| Reward Model | 38.2 (60.0) | 53.8 (-7.8) | 70.6 (35.6) | 45.6 (76.1) | 55.4 (3.4) | 66.2 (30.2) | 74.2 (28.2) | 79.2 (88.2) |
| **NTF (Ours)** | **43.0** (120.0) | **68.6** (107.8) | **78.8** (91.8) | **47.8** (100.0) | **61.2** (103.4) | **73.0** (94.3) | **79.8** (100.0) | **80.4** (105.9) |
| Ground Truth | 41.4 | 67.6 | 80.0 | 47.8 | 61.0 | 73.6 | 79.8 | 80.0 |

*Table 26.* **Individual results for OPENBOOKQA.** Accuracy (%) is reported for each teacher–student setting. Recovery (Eq. 1) for each baseline is reported inside parentheses.

| Teacher → | OLMo2-1B | | | Qwen3-0.6B | | | | |
|---|---|---|---|---|---|---|---|---|
| Method ↓ – Student → | OLMo2-1B | OLMo2-7B | OLMo2-13B | Qwen3-0.6B | Qwen3-1.7B | Qwen3-4B | Qwen3-8B | Qwen3-14B |
| Teacher Performance | 60.3 | | | 71.3 | | | | |
| No-SFT | 60.3 | 83.2 | 86.6 | 71.3 | 86.7 | 86.4 | 91.0 | 93.0 |
| Naive | 68.2 (95.2) | 86.7 (63.6) | 90.8 (102.4) | 75.5 (60.9) | 87.7 (52.6) | 89.7 (80.5) | 93.0 (55.6) | 94.7 (54.8) |
| I-Confidence | 68.4 (97.6) | 85.5 (41.8) | 91.0 (107.3) | 75.6 (62.3) | 88.4 (89.5) | **90.8** (107.3) | 92.9 (52.8) | 94.7 (54.8) |
| ICL + I-Confidence | 69.1 (106.0) | 86.9 (67.3) | 90.7 (100.0) | 76.8 (79.7) | 89.1 (126.3) | 90.4 (97.6) | 93.7 (75.0) | 94.7 (54.8) |
| Ensemble | 66.2 (71.1) | 86.3 (56.4) | 90.6 (97.6) | 75.6 (62.3) | 88.6 (100.0) | 89.1 (65.9) | 93.2 (61.1) | 94.3 (41.9) |
| Reward Model | 59.4 (-10.8) | 84.2 (18.2) | 90.7 (100.0) | 71.7 (5.8) | 84.1 (-136.8) | 87.7 (31.7) | 93.3 (63.9) | **95.4** (77.4) |
| **NTF (Ours)** | **69.6** (112.0) | **87.5** (78.2) | **92.5** (143.9) | **77.0** (82.6) | **89.5** (147.4) | 89.9 (85.4) | **93.8** (77.8) | 95.0 (64.5) |
| Ground Truth | 68.6 | 88.7 | 90.7 | 78.2 | 88.6 | 90.5 | 94.6 | 96.1 |

*Table 27.* **Individual results for SCIQ.** Accuracy (%) is reported for each teacher–student setting. Recovery (Eq. 1) for each baseline is reported inside parentheses.

| Teacher → | **OLMo2-1B** | | | **Qwen3-0.6B** | | | | |
|---|---|---|---|---|---|---|---|---|
| **Method ↓ – Student →** | OLMo2-1B | OLMo2-7B | OLMo2-13B | Qwen3-0.6B | Qwen3-1.7B | Qwen3-4B | Qwen3-8B | Qwen3-14B |
| Teacher Performance | | **38.5** | | | | **43.6** | | |
| No-SFT | 38.5 | 51.7 | 62.2 | 43.6 | 53.1 | 52.5 | 54.2 | 55.8 |
| Naive | 46.5 (109.6) | 61.2 (88.0) | 67.0 (59.3) | 52.6 (93.8) | 62.4 (87.7) | 61.6 (74.0) | 67.1 (86.0) | 72.2 (93.2) |
| I-Confidence | 46.8 (113.7) | 61.8 (93.5) | 67.5 (65.4) | 52.8 (95.8) | 62.7 (90.6) | 62.4 (80.5) | 68.3 (94.0) | 71.6 (89.8) |
| ICL + I-Confidence | 47.7 (126.0) | **62.9 (103.7)** | 68.4 (76.5) | **54.7 (115.6)** | 62.6 (89.6) | 62.4 (80.5) | **70.2 (106.7)** | 73.2 (98.9) |
| Ensemble | 47.0 (116.4) | 60.6 (82.4) | 67.6 (66.7) | 52.3 (90.6) | 62.1 (84.9) | 61.1 (69.9) | 67.8 (90.7) | 71.3 (88.1) |
| Reward Model | 41.4 (39.7) | 59.6 (73.1) | **68.7 (80.2)** | 50.1 (67.7) | 57.8 (44.3) | 58.7 (50.4) | 64.8 (70.7) | 70.5 (83.5) |
| **NTF (Ours)** | **48.0 (130.1)** | 62.7 (101.9) | 68.1 (72.8) | 53.3 (101.0) | **63.1 (94.3)** | **63.7 (91.1)** | 68.9 (98.0) | **74.7 (107.4)** |
| Ground Truth | 45.8 | 62.5 | 70.3 | 53.2 | 63.7 | 64.8 | 69.2 | 73.4 |

*Table 28.* **Individual results for SOCIALIQA.** Accuracy (%) is reported for each teacher–student setting. Recovery (Eq. 1) for each baseline is reported inside parentheses.

