# OpenReview forum: "Trust Functions: Near Lossless Weak-to-Strong Generalization by Learning to Trust the Weak Teacher"
_ICML.cc/2026/Conference — ICML 2026 regular_

### Official Review · Reviewer_vtiS · 2026-03-10

**Soundness:** 2
**Presentation:** 3
**Significance:** 2
**Originality:** 2
**Overall Recommendation:** 3
**Confidence:** 4

**Summary:**

The paper proposes trust function that assign scaler trust score for weak labels. Trust functions enable an iterative chain by training a student and reusing it as the next teacher, producing the strongest final model.

**Compliance With Llm Reviewing Policy:**

Affirmed.

**Key Questions For Authors:**

- How does the model determine the threshold for filtering weak labels?
- What is the computational complexity of training NTFs with the teacher model?
- How did you select the n training examples?
- Neural trust and activation functions are not well explained in the paper
- Correctness is improved with depth, late layers achieve better results is basis of deep learning models.
- How do you think the results will be different using attention or mean pooling methods?
- Why did you use only the hidden state for the final output rather than aggregation multiple tokens?
- Why do deeper layers achieve better AUC but worse calibration compared to layer 25?

**Limitations:**

There is limitation section in the paper.

**Strengths And Weaknesses:**

Although the idea is not novel, training a stronger model using supervision from a weaker model, the performance of NTF is better in most cases. The proposed method uses weak labels generated by a teacher model rather than expensive annotation process. The results are close to ground truth in most settings.

---

> ### Author Rebuttal · Authors · 2026-03-31
>
> We thank the reviewer for recognizing that NTF outperforms baselines in most cases, achieves results close to ground truth in most settings, and avoids expensive annotation by leveraging weak teacher labels. We address the remaining concerns below.
>
> > **Q1 & Q3**
>
> Our framework does not require a manually set threshold. As detailed in App. D, the supervision budget n is set to match the number of correct weak labels in the pool, ensuring all methods operate under a matched budget so that performance differences directly reflect selection quality.
>
> For realistic settings that lack ground-truth labels, App. I introduce a calibration procedure that estimates n from a small labeled subset, achieving high purity with well-characterized variance. We further extend our calibration procedure with a risk-control methodology [here](https://openreview.net/forum?id=6bXLKr5NMT&noteId=5iAEWkVxQi), which guarantees a user-specified purity for the target domain we want to train students for.
>
> > **Q2**
>
> NTFs are lightweight to train, and we provide a detailed cost analysis below.
>
> Let $N_u$ denote the size of the unlabeled target pool, $N_\ell \ll N_u$ the labeled source set, $C_{\text{teacher}}$ the cost of a single teacher forward pass, $D$ the teacher hidden dimension, $W$ the NTF width, $L$ the NTF depth, and $E$ the number of training epochs. The total cost decomposes into two terms, teacher hidden state extraction and NTF training:
> $$C_{\text{total}} = O(N_u \cdot C_{\text{teacher}}) + O(N_\ell \cdot E \cdot (D \cdot W + L \cdot W^2))$$
>
> Since hidden state extraction can be performed jointly with weak label generation, it introduces no additional teacher queries. The NTF training term is negligible by three compounding factors: (i) $N_\ell \ll N_u$, with Table 10 showing strong performance at $N_\ell = 1{,}000$; (ii) $W \in [512, 1024]$ and $L \in [4, 8]$ (Table 13) are orders of magnitude smaller than the teacher; and (iii) $E$ is small due to the lightweight nature of the NTF. Thus:
>
> $$C_{\text{total}} \approx O(N_u \cdot C_{\text{teacher}})$$
>
> Therefore, the cost of training NTFs is dominated by the cost of generating weak labels.
>
> > **Q4**
>
> NTF is a learnable neural network that predicts whether a weak teacher’s label is correct from the teacher’s hidden states. We use the last-layer hidden state of the final generated token as input and a small residual MLP with RMSNorm-SwiGLU blocks, followed by a sigmoid head, to output a scalar trust score. We will clarify this in Sec. 2.2 and to connect it to the ablations on layer choice, token position, and pooling.
>
> > **Q5**
>
> We agree that later layers often contain more task-relevant information. We therefore study layer choice in App. A.1 and select deeper layers based on empirical performance. We will clarify this motivation in the paper.
>
> > **Q6 & 7**
>
> We did not fix the representation choice a priori. As described in the App. A, we ablate layer, token position, and token aggregation. We find that the final output token clearly outperforms the last input token, and that mean pooling over output tokens is worse than using the final output token alone.
>
> We also tested an attention-based pooling baseline over last-layer output tokens. Given output hidden states $h_1,\dots,h_n$, it computes scores $e_t = v^\top \tanh(Wh_t)$, normalizes them with softmax to obtain $\alpha_t$, and forms a weighted sum $z=\sum_i \alpha_i h_i$, which is then fed to the NTF. Although this lets the model emphasize potentially informative positions, it still performs substantially worse than our chosen representation, achieving 0.8656 AUC, 0.0569 ECE, and 0.1444 Brier.
>
> Thus, we use the final output token hidden state from the last layer in the main experiments.
>
> > **Q8**
>
> AUC measures discrimination, while ECE measures calibration. Thus, a deeper layer can have a higher AUC if it better separates correct from incorrect cases, yet worse ECE if its confidence scores are less aligned with true probabilities. In our results, however, the difference between layers 25 and 28 is very small, since layer 28 has slightly higher AUC, while layer 25 has slightly lower ECE. So, the two layers perform similarly overall.
>
> This pattern is also consistent with prior work. In [1, Fig. 1], predictive accuracy improves and then saturates across depth, while ECE continues to vary, suggesting calibration is not monotonic across layers. Similarly, [2, Fig. 2] shows that deeper representations can yield higher linear-probe accuracy while becoming less well calibrated. Together, these observations support our interpretation that later layers may be slightly more discriminative but not necessarily better calibrated.
>
> > **Summary**
>
> NTF needs no manual threshold, is lightweight, and its design choices are empirically validated.
>
> [1] Joshi, A., et al. Calibration Across Layers: Understanding Calibration Evolution in LLMs.
>
> [2] Wang, X., et al. Calibration Bottleneck: Over-compressed Representations are Less Calibratable.

---

> > ### Author Rebuttal · Reviewer_vtiS · 2026-03-31
> >
> > The authors have responded to all comments.

---

> > > ### Author Response · Authors · 2026-04-03
> > >
> > > Thank you again for the acknowledgement and for noting that our rebuttal fully resolved your concerns. We noticed that the score may not have been updated alongside that acknowledgement. If your current assessment is indeed that the concerns have been resolved, we would be very grateful if you could reconsider the score to reflect that view. In any case, we appreciate your time and feedback.

---

### Official Review · Reviewer_DATR · 2026-03-12

**Soundness:** 3
**Presentation:** 4
**Significance:** 3
**Originality:** 4
**Overall Recommendation:** 5
**Confidence:** 4

**Summary:**

The paper studies weak-to-strong generalization as a data selection problem where instead of using all weak labels, reliable ones are identified. The authors propose to train a small probe or NTF (neural trust function) trained on hidden stated from a teacher to filter weak labels before student training. NTFs demonstrate gains across three substantially different domains where students trained on trust filtered weak labels are at par with the models trained on ground truth labels and in several cases, surpass them.

**Compliance With Llm Reviewing Policy:**

Affirmed.

**Final Justification:**

I have read the author's rebuttal carefully and will maintain the positive score.

**Key Questions For Authors:**

1. Can you provide the results for a vanilla chaining baseline (iterative training on Naive/unfiltered labels) for the experiments in Section 5?
2. Why does the method work significantly better for Qwen3 model family than OLMo2? Is there a model bias or are certain family representations better suited for NTF?

**Limitations:**

Yes

**Strengths And Weaknesses:**

**Strengths**
1. The motivation is well grounded and the paper is very well framed. The comparison between output level heuristics and representation level trust is empirically demonstrated very well.
2. The experimental scope of the paper is broad and covers three tasks across two model families and range of scales/size of the model. The near lossless performance recovery is validated and tested for significance.
3. Section 5 shows that the final Qwen3-14B student, trained through a chain of trust-filtered students-as-teachers, performs better than both a single-step transfer from the strongest available weak teacher and ground-truth supervised training under the same budget which is an important result.
4. The ablations are very well structured and address all design questions such as which transformer layer to probe, which token position to use, whether to pool across tokens, how much labeled training data the NTF needs, etc.

**Weaknesses**
1. The practical scope of applicability could be strengthened if the authors ablate the minimum labeled source set size needed for NTF to outperform confidence-based baselines.
2. The analysis also lacks a vanilla chaining baseline where students are trained on unfiltered weak labels across iterations to understand if compounding gains come from trust filtering specifically or simply from having a better teacher at each step.

---

> ### Author Rebuttal · Authors · 2026-03-31
>
> We thank the reviewer for the detailed and insightful review and strong overall assessment of our work. We are encouraged that the reviewer found our motivation "well grounded" and the paper "very well framed," and appreciated the empirical demonstration comparing output-level heuristics to representation-level trust. We are also glad the reviewer recognized the broad experimental scope across tasks, model families, and scales, and found the significance testing for near-lossless recovery to be well-validated. We particularly appreciate the reviewer highlighting the chaining result in Section 5 that the final Qwen3-14B student outperforms both single-step transfer from the strongest available weak teacher and ground-truth supervised training as "an important result," as well as their assessment that our ablations "address all design questions." We respond to the reviewer's remaining questions below.
>
> > **W1: The minimum labeled source set size needed for NTF to outperform confidence-based baselines.**
>
> We thank the reviewer for highlighting this. To address this concern, we add a new ablation in the decision-making domain, where we vary the number of training examples picked by NTF and GT baselines. We compare against the GT baseline since it is a stronger reference point, and if NTF matches or outperforms GT, it is already operating in a regime stronger than confidence-based filtering.
>
> | $n$ | Qwen3-1.7B (NTF) | Qwen3-1.7B (GT) | Qwen3-4B (NTF) | Qwen3-4B (GT) |
> |---|---:|---:|---:|---:|
> | 500 | 0.00 | 0.00 | 0.01 | 0.01 |
> | 1000 | 0.24 | 0.21 | 1.74 | 1.60 |
> | 5000 | 3.44 | 1.66 | 5.75 | 4.02 |
> | 10000 | 17.11 | 18.62 | 30.16 | 28.35 |
> | 25000 | 27.41 | 23.79 | 33.50 | 38.67 |
> | 50000 | 25.40 | 23.00 | 35.40 | 36.30 |
>
> As shown in the table, NTF outperforms GT in 8 out of 12 model-$n$ combinations. Overall, the ablation indicates that NTF becomes competitive well before the largest set sizes and often surpasses the matched GT baseline across a broad range of data budgets.
>
> > **W2/Q1: Vanilla chaining baseline for the experiments in Sec. 5.**
>
> We thank the reviewer for highlighting this important baseline. We now include a naive chaining baseline, where the model is iteratively trained on naive labels.
>
> | Method | Qwen3-4B | Qwen3-8B | Qwen3-14B |
> | :--- | :---: | :---: | :---: |
> | **NTF Chain** | 36.9 | 40.1 | 48.2 |
> | **NTF Shallow (0.6B)** | 35.4 | 38.0 | 44.1 |
> | **NTF Shallow (8B)** | — | — | 46.1 |
> | **GT** | 36.2 | 37.0 | 40.0 |
> | **Naive Chain** | 30.1 | 34.2 | 39.1 |
> | **Naive Shallow (0.6B)** | 27.0 | 33.7 | 38.1 |
>
> As the table suggests, the naive chaining outperforms single-round (shallow) training on naive labels; yet, it lags behind single-round and chained NTF training as well as single-round GT training. This demonstrates that the snowballing gains in Section 5 are not explained by chaining alone, but specifically by the trust-filtered supervision used at each stage.
>
> > **Q2: Model family bias.**
>
> We also noticed that the gain over baselines is more pronounced for the Qwen3 family than for OLMo2 in the decision-making domain. However, Appendix A.5 suggests that this difference is unlikely to be driven by a failure of the NTF itself. In Table 12, the NTF for OLMo2-1B teacher achieves strong discrimination and calibration on Lichess_{train} (AUC 0.930, ECE 0.017, Brier 0.086, Purity 0.930), which is comparable to the NTF for Qwen3-0.6B teacher (AUC 0.914, ECE 0.022, Brier 0.113, Purity 0.953). This suggests that the trust-based data selection is functioning well for both families, and that the remaining difference may be due to family-specific representation or optimization effects in downstream chess learning rather than NTF performance itself. Importantly, we do not observe the same pattern in the MCQA setting, where both OLMo2 and Qwen3 show near lossless weak-to-strong recovery and strong NTF evaluation metrics. We will clarify this in the revision.
>
> > **Additional Comments:**
>
> We’d like to show our results on additional results on data selection mentioned by Reviewer q9Mk [here](https://openreview.net/forum?id=6bXLKr5NMT&noteId=ipzhaZqsrm) to strengthen the practical use case of NTF and the Learning-to-Trust pipeline.
>
> > **Summary**
>
> We will revise the draft in three ways. First, we will add a naive chaining baseline in Section 5, showing that iterative training on unfiltered labels helps over shallow naive training but remains below shallow and chained NTF training, as well as shallow GT training. Second, we will clarify that the Qwen3-OLMo2 gap in chess is unlikely to reflect NTF failure, but instead likely arises from family-specific downstream learning dynamics. Third, we will add a new ablation on the data amount showing that NTF outperforms the GT baseline in 8 of 12 settings across different amounts of data.

---

> > ### Author Rebuttal · Reviewer_DATR · 2026-04-01
> >
> > I have read the author's rebuttal carefully and will maintain the positive score.

---

### Official Review · Reviewer_q9Mk · 2026-03-13

**Soundness:** 3
**Presentation:** 3
**Significance:** 3
**Originality:** 3
**Overall Recommendation:** 5
**Confidence:** 3

**Summary:**

This paper introduces trust functions to improve weak-to-strong generalization by identifying and filtering reliable training signals from weak teacher models. By assigning scalar trust scores to teacher labels, the method achieves near-lossless performance across diverse domains, often matching or exceeding the accuracy of models trained on ground-truth supervision. The authors further demonstrate that this approach enables an iterative learning chain and induces a beneficial easy-first training curriculum that stabilizes gradient updates.

**Compliance With Llm Reviewing Policy:**

Affirmed.

**Key Questions For Authors:**

See weaknesses section.

**Limitations:**

Yes

**Strengths And Weaknesses:**

Strengths:
- The trust function filtering allows student models to achieve near-lossless generalization, frequently matching or exceeding the accuracy of models trained on actual ground-truth labels.
- The approach demonstrates strong results across a wide range of tasks, including world knowledge, quantitative reasoning, and decision making, proving it is not limited to a single type of data.
- The introduction of a weak-to-strong chain allows for compounded gains, where a trained student becomes a more effective teacher for the next generation of models.
- The paper provides a mechanistic analysis showing that trust filtering stabilizes training by creating an implicit easy-first curriculum and ensuring more coherent gradient updates.
- The setup and results showing that the trust function is trained on easier source datasets but transfers to target data are interesting and practically meaningful

Weaknesses:
- The effectiveness of the near-lossless generalization likely depends on careful tuning of the trust score thresholds, and the paper provides limited guidance on how these thresholds behave across different datasets.

---

> ### Author Rebuttal · Authors · 2026-03-31
>
> We thank the reviewer for the careful reading and positive assessment of our work. We are glad the reviewer recognized the strength of our near-lossless generalization results, the breadth of evaluation, and the practical significance of the iterative weak-to-strong chain. We especially appreciate the reviewer’s observation that “the setup and results showing that the trust function is trained on easier source datasets but transfers to target data are interesting and practically meaningful,” as this zero-shot transfer property is central to the practical utility of our framework. We are grateful for the reviewer’s recommendation to accept and address the remaining concern below.
>
> > **W1: Threshold selection.**
>
> We elaborate how we determine the number of training examples in App. D. We also do a calibration analysis on how to use NTFs in App. I. To further address this concern, we now provide i) an empirical analysis on the calibration of our data selection used in the paper, ii) a new statistical method to achieve user-specified high-purity with theoretical guarantees.
>
> First, we want to emphasize our data selection method, as described in the App. D, is retaining top-n highest trust examples in the pool of weak labels. Nonetheless, we provide empirical analyses of trust score threshold across datasets ([[a]](https://anonymous.4open.science/r/ICML_rebuttal_figures-A644/NTF_score_distribution/), [[b]](https://anonymous.4open.science/r/ICML_rebuttal_figures-A644/NTF_threshold_curves/)). These results show that the trust scores are well calibrated and that a threshold of 0.8 consistently gives a strong tradeoff between purity and retained volume.
>
> Second, beyond this empirical guidance, we provide a more principled risk-controlled data selection procedure, inspired by Learn-Then-Test [1]. The goal is to select a threshold using a small labeled calibration subset from the target OOD pool (i.e., the target domain we want to train students for), while providing a statistical guarantee on the noise rate of the selected data. Let $s((x,\hat{y}))$ in $[0,1]$ denote the trust score for an input-prediction pair $(x, \hat{y})$, where larger scores indicate higher confidence that the weak label is trustworthy.
>
> For a candidate threshold $\tau$, we define the selected set as
> $S(\tau) = $ { $(x,\hat{y}) : s((x,\hat{y})) \geq \tau $ }.
>
> On a labeled calibration subset, we estimate the empirical noise rate of this rule as
> $\hat{r}(\tau) = \frac{1}{n(\tau)} \sum_{i : s_i \geq \tau} (1 - y_i)$,
> where $y_i = 1$ means the prediction is correct, and $n(\tau)$ is the number of calibration examples with $s_i >= \tau$.
>
> To obtain a high-confidence guarantee, we do not select thresholds using empirical noise alone. Instead, for each candidate threshold we compute an upper confidence bound (UCB) on the true noise rate. With the Hoeffding bound [2], this is
> $U(\tau) = \hat{r}(\tau) + \sqrt{\frac{\log(1 / \delta_{\tau})}{2 n(\tau)}}$.
>
> We then select the most inclusive threshold satisfying
> $U(\tau) \leq \alpha$,
> where $\alpha$ is a user-specified target noise rate. This yields the guarantee that, with probability at least $1 - \delta$, the selected rule has a true noise rate at most $\alpha$.
>
> When the threshold is chosen adaptively from multiple candidates using the same calibration set, we apply a Bonferroni correction [3]. If $m$ candidate thresholds are evaluated, we use
> $\delta_{\tau} = \frac{\delta}{m}$.
>
> This ensures that the final selected threshold still satisfies the desired confidence guarantee despite searching over multiple threshold values. Intuitively, this procedure chooses the largest retained subset whose worst-case plausible noise rate is still below the user’s target.
>
> In our implementation, we consider two related selection modes. In the threshold mode, we calibrate the trust score threshold $\tau$ directly. In the top-$k$ mode, we instead sort examples by trust scores and choose the largest $k$ whose UCB on noise is below $\alpha$. This can be viewed as calibrating how much data to keep rather than calibrating a score threshold directly. The top-$k$ solution also induces a corresponding confidence threshold through the score of the $k$-th selected item.
>
> Empirically, these risk-control experiments support the same conclusion as our descriptive threshold sweeps, where a threshold around 0.8 is a strong operating point ([[c]](https://anonymous.4open.science/r/ICML_rebuttal_figures-A644/NTF_risk_control_data_selection_chess/)). Thus, our revised paper will provide both practical guidance (0.8 is a robust default operating point) and a principled calibration procedure for cases where users want a formally controlled noise level on a new dataset.
>
> [1] Angelopoulos, et al. (2021). Learn then test: Calibrating predictive algorithms to achieve risk control.
>
> [2] Hoeffding, et al. (1963). Probability inequalities for sums of bounded random variables.
>
> [3] Dunn, et al. (1961). Multiple comparisons among means.

---

> > ### Author Rebuttal · Reviewer_q9Mk · 2026-04-03
> >
> > Thanks for the response. I am satisfied with the authors’ responses and will keep my positive score.

---

### Official Review · Reviewer_BYKu · 2026-03-13

**Soundness:** 4
**Presentation:** 4
**Significance:** 4
**Originality:** 2
**Overall Recommendation:** 5
**Confidence:** 3

**Summary:**

- The paper proposes a framework to improve the performance of weak-to-strong generalization by first training a trust function that determines whether a weak label is correct or not and then using the learned trust function to filter the training data.
- The paper provides experiments on multiple domains, from world knowledge to reasoning. The results show that the proposed method performs as well as using ground truth labels.
- The paper also suggests that the framework is suitable for iterative training.
- Finally, the paper provides ablation studies of the distribution of the selected samples and gradients to improve the understanding of why the trust function works.

**Compliance With Llm Reviewing Policy:**

Affirmed.

**Final Justification:**

The rebuttals have addressed my question regarding the experiments. Therefore, I have increased the score.

**Key Questions For Authors:**

- We can replace the trust function with any off-the-shelf reward model. How would NFT compare to this baseline? It would be interesting if one could show that training a neural trust function is necessary in some aspect.
- Is there any instance where the NTF is confident but wrong? How to mitigate this ?
- What is the performance of NFT as n changes? I doubt that NFT would achieve lossless performance to ground truth with a larger n since NFT prefers easy examples.

**Limitations:**

yes

**Strengths And Weaknesses:**

**Soundness:**
- The experiment result is quite strong. The proposed method (NTF) performs as well as using the ground truth label on many tasks.
- The paper also demonstrates that this approach works for iterative training which is nice.
- Ablation studies in section 6 is comprehensive and do help explain the mechanism behind why the NTF works quite well.

**Presentation:**
- Well-written and is easy to follow.
- The experiment settings and results are clear.

**Significance:**
- The problem studied here is important and interesting.

**Originality:**
- The idea of data selection for weak supervision is not novel. This has been explored in the weak supervision literature [1][2], especially [1].
- Further, the trust function resembles a verifier function that is used to evaluate the quality of an output of a model [3].
- It would be nice if the paper cites these prior works and discusses them.

[1] Dehghani, Mostafa, et al. "Learning to learn from weak supervision by full supervision." arXiv preprint arXiv:1711.11383 (2017).
[2] Lang, Hunter, Aravindan Vijayaraghavan, and David Sontag. "Training subset selection for weak supervision." Advances in Neural Information Processing Systems 35 (2022): 16023-16036.

[3] Cobbe, Karl, et al. "Training verifiers to solve math word problems." arXiv preprint arXiv:2110.14168 (2021).


Overall, while the idea is not as novel. I appreciate the effort in applying this idea with the recent LLMs training recipe (SFT, GRPO) and also the additional ablations.

---

> ### Author Rebuttal · Authors · 2026-03-31
>
> We thank the reviewer for the thorough and constructive review. We appreciate the reviewer’s positive assessment of our empirical results, iterative training analysis, and mechanistic ablations, and we address the questions below.
>
> > **Originality: Prior work.**
>
> We agree that the general idea of improving learning from imperfect supervision via data selection is not entirely new, and we will revise the paper to make this positioning explicit.
>
> Prior work [1,2] focuses on in-domain weakly supervised learning, whereas our paper studies weak-to-strong generalization (W2SG) under distribution shift. Notably, our pipeline admits any prevalent learning algorithm (e.g., SFT, RL), whereas [1] is limited by the learning algorithm they propose. [2] depends on student representations whose neighborhood geometry is expected to align with weak-label quality. By contrast, our setting estimates the reliability of weak labels for W2SG, and it is unclear that such geometry captures this signal, especially under distribution shift. Lastly, [1, 2] study non-reasoning tasks.
>
> We also agree that our trust function is conceptually related to a verifier [3], which we cite in App. C. However, there are two key differences: 1) the reward model works on textual input, while our trust functions work on hidden representations, and 2) the training cost of a separate reward model is substantially higher than the training cost of NTFs, which is dominated by weak label generations, as shown [here,Q2](https://openreview.net/forum?id=6bXLKr5NMT&noteId=JjC4HGpeNW).
>
> > **Q1: Using reward models.**
>
> We agree this is an important baseline, and we add experiments replacing NTF with public reward models. For math, we use Qwen2.5-Math-RM-72B [1]. For world knowledge, we test ArmoRM-Llama3-8B-v0.1 [2], Skywork-Reward-V2-Qwen3-8B [3], and CompassVerifier-3B [4], selecting the highest-purity RM for student training in each benchmark.
>
> Results are shown in the [table 1](https://anonymous.4open.science/r/ICML_rebuttal_figures-A644/MCQA_RM_results.png
> ) and [table 2](https://anonymous.4open.science/r/ICML_rebuttal_figures-A644/Math_RM_results.png
> ). In the world knowledge domain, NTF always outperforms RM, often by a substantial margin. In math, NTF is comparable to RM, suggesting that large and specialized RMs can be competitive. We view this as a strong result for NTF rather than a weakness, since Qwen2.5-Math-RM-72B is a large math-specialized RM, and our math setting may also benefit RM due to the distributional proximity between Omni-MATH [5] and public math corpora such as NuminaMath [6], which is used to train Qwen2.5-Math-RM-72B.
>
> Lastly, our contribution is the learning-to-trust framework, with NTF as one concrete instantiation. Stronger future trust estimators, including improved reward models, are complementary to our framework.
>
> > **Q2: NTF being confident but wrong.**
>
> We find that confident incorrect NTF predictions are rare. As shown in Table 12, the NTF is already well calibrated. Consequently, the resulting noise is limited and does not appear to meaningfully affect training. We also provide additional calibration analyses and a risk-controlled data-selection pipeline [here](https://openreview.net/forum?id=6bXLKr5NMT&noteId=ipzhaZqsrm), which further reduces the chance of selecting confidently incorrect examples. Thus, we leave improving robustness in these rare failure cases as valuable future work.
>
> > **Q3: Performance as n changes.**
>
> In our original experiments, NTF already uses roughly the largest feasible n (rounded down to 50k), because it is constrained by the number of correct answers produced by the weak teacher. Increasing n further would necessarily reduce purity by introducing incorrect weak labels. We also note that performance does not consistently improve as more ground-truth data are added, so we did not test n>50k.
>
> However, we agree this is an important ablation. We therefore ran additional decision-making experiments with varying n. As shown in the [table](https://anonymous.4open.science/r/ICML_rebuttal_figures-A644/chess_n_ablation.png), NTF remains competitive with, and often outperforms, GT (8 of 12 model-n combinations), although GT can be stronger at larger n.
>
> > **Summary**
>
> We add RM baselines, conduct ablation on training dataset size (n), expand NTF calibration analysis, and clarify the positioning of originality.
>
> [1] Yang, et al. (2024). Qwen2.5-Math technical report: Toward mathematical expert model via self-improvement.
> [2] Wang, et al. (2024). Interpretable preferences via multi-objective reward modeling and mixture-of-experts.
> [3] Liu, et al. (2026) Skywork-Reward-V2: Scaling Preference Data Curation via Human-AI Synergy.
> [4] Liu, et al. (2025). CompassVerifier: A Unified and Robust Verifier for LLMs Evaluation and Outcome Reward.
> [5] Gao, et al. (2024). Omni-MATH: A universal olympiad level mathematic benchmark for large language models
> [6] Li, et al. (2024). NuminaMath Dataset. Hugging Face.

---

> > ### Author Rebuttal · Reviewer_BYKu · 2026-04-01
> >
> > The rebuttals have addressed my questions. I am increasing my score.

---

### Decision · Program_Chairs · 2026-04-30

**Decision:**

Accept (regular)

**Comment:**

The paper proposes trust functions for filtering weak labels in weak-to-strong generalization. It demonstrated near-lossless performance across multiple domains, with iterative chaining for further improvement. Reviewers appreciate the paper's empirical results, evaluation across domains and model families, and the fact that the method often matches or even surpasses ground-truth supervision. All reviewers stated in their rebuttal acknowledgements that their concerns were adequately addressed or fully resolved, including reviewer vtiS whose score remained at 3, so I recommend accept.